# UEPI: Universal Energy-Behavior-Preserving Integrators for Energy Conservative/Dissipative Differential Equations

**Elena Celledoni**
Department of Mathematical Sciences
Norwegian University of Science and Technology
Trondheim, Norway
elena.celledoni@ntnu.no

**Brynjulf Owren**
Department of Mathematical Sciences
Norwegian University of Science and Technology
Trondheim, Norway
brynjulf.owren@ntnu.no

**Chong Shen**
Graduate School of Science
Kobe University
Kobe, Japan
242s026s@stu.kobe-u.ac.jp

**Baige Xu**
Graduate School of Science
Kobe University
Kobe, Japan
baigexu@stu.kobe-u.ac.jp

**Takaharu Yaguchi**
Graduate School of Science
Kobe University
RIKEN AIP
Kobe, Japan
yaguchi@pearl.kobe-u.ac.jp

## Abstract

Physical phenomena in the real world are often described by energy-based modeling theories, such as Hamiltonian mechanics or the Landau theory. It is known that physical phenomena based on these theories have an energy conservation law or a dissipation law. Therefore, in the simulations of such physical phenomena, numerical methods that preserve the energy-conservation or dissipation laws are desirable. However, because various energy-behavior-preserving numerical methods have been proposed, it is difficult to discover the best one. In this study, we propose a method for learning highly accurate energy-behavior-preserving integrators from data. Numerical results show that our approach certainly learns energy-behavior-preserving numerical methods that are more accurate than existing numerical methods for various differential equations, including chaotic Hamiltonian systems, dissipative systems, and a nonlinear partial differential equation. We also provide universal approximation theorems for the proposed approach.

## 1 Introduction

Differential equations serve as fundamental tools for modeling the dynamics of various physical systems. However, most real-world differential equations cannot be solved analytically, necessitating the use of numerical methods to obtain approximate solutions. A critical challenge in this context is ensuring both accuracy and stability over long simulation times, especially for systems governed by complex or nonlinear dynamics.

39th Conference on Neural Information Processing Systems (NeurIPS 2025).

Table 1: Comparison of the proposed method, discrete gradients, SympNets [31]. Because the proposed method is a numerical integrator, the approximation accuracy is always guaranteed to a certain extent, regardless of the results of training.

|  | naive Runge-Kutta | discrete gradient | SympNets | Proposed |
|---|:---:|:---:|:---:|:---:|
| numerical integrators | ✓ | ✓ |  | ✓ |
| energy conservation |  | ✓ | ✓ | ✓ |
| energy dissipation |  | ✓ |  | ✓ |
| learnable from data |  |  | ✓ | ✓ |

In recent years, advances in machine learning have inspired a new class of **data-driven numerical methods** [1, 12, 21], where neural networks are used to optimize the coefficients or design high-order schemes for classical integrators such as Runge-Kutta methods. These methods offer remarkable adaptability and high short-term accuracy when tailored to specific problems. However, a significant limitation is their lack of preservation of key invariants, such as energy conservation or symplecticity, which are essential for long-term simulations. In contrast, geometric numerical integrators [24] are specifically designed to preserve such intrinsic structures, thereby ensuring better performance over long time intervals. Among them, energy-preserving methods such as the discrete gradient methods[19, 47] have gained attention for their ability to conserve the system's energy evolution precisely. To the best of our knowledge, existing literature has not yet explored the potential of neural networks in optimizing energy-preserving geometric integrators.

In this study, we propose a method for learning numerical integrators that preserve the energy behaviors of target differential equations. More specifically, the focus of this paper is on dynamical systems characterized by their energy-conserving or dissipating properties. We present a general form of the discrete gradient method and propose a method for learning highly accurate energy-behavior-preserving integrators from data by employing neural networks. In addition, we show universal approximation theorems for the proposed approach.

The main contributions of this paper are as follows.

- We characterize and then parameterize the energy-behavior-preserving numerical integrators.

- We propose neural network models with universal approximation properties for the energy-behavior-preserving numerical integrators. Numerical experiments have confirmed that the method can be applied to a wide range of phenomena, including chaotic conservative systems, dissipative systems, and nonlinear partial differential equations.

- Numerical experiments also show that the proposed method, in fact, finds highly accurate computational methods, which are difficult to discover theoretically. In particular, it is numerically shown that numerical schemes derived by the proposed method are highly accurate and stable even for chaotic systems.

## 2   Related Work

**Neural networks for improving numerical methods** represent a hybrid paradigm. One line of research focuses on optimizing classical numerical methods through data-driven parameterization rather than approximating dynamics directly. Anastassi [1] developed artificial neural networks(ANNs) to generate the coefficients of two-stage Runge-Kutta methods specifically optimized for solving the two-body problem. Dehghanpour et al. [12] utilized ANNs to automatically compute the optimal coefficients of third-order Runge-Kutta methods. Similarly, Guo et al. [21] explored machine learning methods based on the Runge-Kutta-integrator architecture, capable of automatically generating high-order integrators for specific families of ODEs. Universal Numerical Integrators (UNIs) [60, 61] proposed a framework, which combines neural networks as universal approximators with numerical integrators. A second distinct line of work investigates learning discretization schemes from data. Bar-Sinai et al. [2] introduced a data-driven discretization approach that employs neural networks to optimize the computation of spatial derivatives. Maddu et al. [44] developed STENCIL-NET, which can adaptively learn local discretization operators based on solution data, addressing long-term prediction challenges for unknown nonlinear PDEs. Ranade et al. [55] introduced DiscretizationNet, a generative CNN-based approach to model discretization schemes. Our approach follows this hybrid

paradigm but shifts the focus from general integrator optimization to preserving the energy behavior of dynamical systems. We achieve this by combining neural networks with the structure-preserving discrete gradient method, rather than modifying traditional integration integrators.

**Geometric numerical integration** focuses on designing numerical methods that preserve certain qualitative characteristics of dynamical systems. Unlike traditional methods such as the explicit Euler method or the classical Runge-Kutta method, which may lead to long-term errors, geometric integrators maintain properties that make them particularly suitable for long-time simulations of physical systems.

Geometric numerical integrators are generally divided into symplectic integrators and energy-preserving integrators, as Zhong [67] and Chartier et al. [6] have shown that it is impossible for a numerical integrator to preserve both properties. Symplectic integrators ([16, 56]) are integrators that preserve the symplectic structure of Hamiltonian systems. Energy-preserving methods aim to conserve the total energy of the system. Among energy-preserving methods, the discrete gradient method([19, 35–37, 47]) is particularly successful and has also been extended to partial differential equations ([5, 46]). Eidnes [15] develop order theory for discrete gradient methods. Projection methods([10, 14, 22, 23]) can also be employed to preserve energy by projecting the numerical solution back onto the manifold that satisfies a certain energy. Norton et al. [50] show that linear projection methods are a subset of discrete gradient methods. Miyatake and Butcher [48] provided a condition to be energy-preserving for continuous stage Runge-Kutta methods. For other geometric numerical integrators, see ([4, 24, 57]) for details. In our study, we place particular emphasis on discrete gradient methods, as they inherently preserve the energy behaviors, making them highly relevant for simulations involving Hamiltonian dynamics and dissipative systems.

**Neural network-Based Physical Modelings** have been widely applied to the study of differential equations by combining data-driven learning with physical principles. Chen et al. [7] introduced Neural ODEs, which model continuous-time dynamics via parameterized vector fields. This idea has been extended in various directions [3, 13, 65]. Another influential framework is that of Physics-Informed Neural Networks (PINNs) [54], which incorporates differential equation residuals into the loss function to enforce physical consistency during training. Numerous extensions have been developed to enhance their accuracy, robustness, efficiency, and scalability [30, 32, 49, 54, 58]. Operator learning [38–42, 64] aims to learn mappings between infinite-dimensional function spaces. Dynamic Mode Decomposition (DMD) and the Koopman operator theory have emerged as powerful data-driven techniques for approximating nonlinear dynamical systems with linear representations [34, 43, 59, 63]. Beyond generic data-driven models, structure-preserving neural networks have been proposed to better reflect the physical properties of dynamical systems. Hamiltonian Neural Networks (HNNs) [20] model the Hamiltonian to enforce energy conservation. Extensions of HNNs [11, 25, 45] have incorporated flexibility and accuracy improvements. Lagrangian Neural Networks (LNNs) [9] take a similar approach via Lagrangian mechanics. Many studies also focus on the symplecticity [8, 31, 51, 62]. For energy-dissipative systems, some studies have introduced neural networks grounded in thermodynamic principles such as the GENERIC framework to ensure stability, interpretability, and physical consistency in modeling irreversible dynamics [26, 27, 66]. Despite their effectiveness in capturing qualitative physical properties, these methods are not numerical integrators. They focus on learning dynamics or flow maps, and their approximation quality can be difficult to control or quantify in a classical numerical analysis sense. This limits their applicability in contexts where rigorous accuracy guarantees or long-term stability are required.

In summary, our method bridges geometric numerical integration and neural networks by learning energy-behavior-preserving integrators. This offers improved long-term stability and accuracy in simulating a wide range of systems.

## 3 Target Energy-Based Dynamical Systems

We focus on differential equations written in the form

$$\dot{u} = S(u)\nabla H(u), \quad u \in \mathbb{R}^d. \tag{1}$$

Here $H \in C^1(\mathbb{R}^d, \mathbb{R})$ and $S \in C(\mathbb{R}^d, \mathbb{R}^{d \times d})$. Differential equations in this form describe various physical systems modeled by ordinary differential equations (ODEs) and discretized partial differential equations (PDEs) [17, 52]. Two classes of problems are usually considered:

**Conservative case:** $S(u)^T = -S(u)$ (skew-symmetric,)

**Dissipative case:** $S(u) \leq O$ (negative semi-definite.)

Note that $S \leq O$ means that the matrix $S$ is negative semi-definite, i.e., $u^\top S u \leq 0$ for any vector $u$. Also, if a matrix $S$ is skew-symmetric, then $u^\top S u = 0$ for any vector $u$.

The systems described by the above equations have the following properties.

**Theorem 3.1.** *The system has the energy dissipation law if $S \leq O$ and the energy conservation law if $S$ is skew-symmetric.*

See Appendix A for the proof. Such differential equations include various equations that appear in physics. For example, if $S$ is skew-symmetric and invertible and also satisfies a certain condition called the Jacobi identity, the equation describes Hamiltonian systems. As the Hamilton equation is a generalization of Newton's equation of motion, mechanical systems that can be described by classical mechanics, such as the motion of a pendulum. As an example outside of physics, the Lotka–Volterra predator–prey model is also a Hamiltonian system. On the other hand, this equation describes dissipative systems when $S$ is negative-definite. A typical example of dissipative systems is the equation of motion for mechanical systems with friction. Thermodynamic systems can often be expressed as dissipative systems as well. Also, this equation has a geometric background; see Appendix B for details.

The system (1) also describes semi-discretized Hamiltonian or dissipative PDEs in space. For example, the PDEs of the form

$$\frac{\partial u}{\partial t} = \frac{\partial}{\partial x}\frac{\delta H}{\delta u} \tag{2}$$

are an example of Hamiltonian PDEs, where $H(u, u_x)$ is the energy density and $\delta H/\delta u$ denotes the variational derivative of $H$, which is defined by $\frac{\delta H}{\delta u} := \frac{\partial H}{\partial u_x} - \frac{\partial}{\partial x}\frac{\partial H}{\partial u}$. This equation admits the energy conservation law: $\frac{\mathrm{d}}{\mathrm{d}t}\int H \mathrm{d}x = 0$ under, for example, the periodic boundary condition. Similarly, the PDEs

$$\frac{\partial u}{\partial t} = \frac{\partial^2}{\partial x^2}\frac{\delta H}{\delta u} \tag{3}$$

admits the energy dissipation property: $\frac{\mathrm{d}}{\mathrm{d}t}\int H \mathrm{d}x \leq 0$ under the periodic boundary condition. See Appendix C for details on the semi-discretization.

Hamiltonian PDEs include the Maxwell equation and the shallow water equations such as the KdV equation, the advection equation, and the Burgers equation. Dissipative PDEs express physical systems derived from the Landau free-energy minimization including the Cahn–Hilliard equation and the phase–field models for phase transitions and pattern formulations. Other target equations include the equations with complex state variables, such as the Schrödinger equation and the Ginzburg–Landau equation. See, e.g., [17] for details.

## 4 Methods

### 4.1 Discrete Gradient Methods as Energy-Behavior-Preserving Integrators

A popular way of discretizing (1) is by a class of numerical schemes called *discrete gradient methods* defined as

$$\frac{u^{(n+1)} - u^{(n)}}{h} = \bar{S}(u^{(n)}, u^{(n+1)})\, \bar{\nabla}H(u^{(n)}, u^{(n+1)}) \tag{4}$$

where $h$ is the step size. In this discrete counterpart to (1) the skew-symmetrix matrix $S(u)$ has been replaced by the matrix $\bar{S}(u, v)$ depending on two arguments, but still required to be skew-symmetric. $\bar{\nabla}H : \mathbb{R}^d \times \mathbb{R}^d \to \mathbb{R}^d$ is the *discrete gradient* defined as follows.

**Definition 4.1.** A discrete gradient for a smooth function $H : \mathbb{R}^d \to \mathbb{R}$ is a continuous mapping $\bar{\nabla}H : \mathbb{R}^d \times \mathbb{R}^d \to \mathbb{R}^d$ that satisfies the following properties:

$$H(v) - H(u) = \langle \bar{\nabla}H(u, v), v - u \rangle \text{ for all } u, v \in \mathbb{R}^d, \tag{5}$$

$$\bar{\nabla}H(u, u) = \nabla H(u) \text{ for all } u \in \mathbb{R}^d \tag{6}$$

The first condition corresponds to the discrete chain-rule $dH(\delta u; u) = \nabla H(u) \cdot \delta u$ for the Fréchet derivative $dH(\cdot; u)$ of $H$ at $u$, where $\delta u$ is an infinitesimal change of $u$. The second condition ensures that a discrete gradient $\bar{\nabla}H$ is certainly an approximation of the gradient $\nabla H$. The inner product is typically the standard Hermitian inner product for ODEs and the discrete $L^2$ inner product for discretized PDEs. For details on discrete gradient methods, see, e.g., [15].

The discrete gradient schemes preserve the energy behavior of the target differential equations in the following sense.

**Theorem 4.2.** *The discrete gradient scheme* (4) *admits the discrete energy conservation law*

$$H(u^{(n+1)}) = H(u^{(n)})$$

*if $\bar{S}$ is skew-symmetric and the discrete energy dissipation law*

$$H(u^{(n+1)}) \leq H(u^{(n)})$$

*if $\bar{S} \leq O$. In particular, if the system is dissipative, the amount of energy dissipation is an approximation of that of the original differential equation.*

*Proof.* The proof is exactly the same as that for Theorem 3.1. See Appendix A for details. □

There are several popular choices for discrete gradients. The average vector field (AVF) discrete gradient [53] is defined as

$$\bar{\nabla}H(u, v) = \int_0^1 \nabla H((1 - \zeta)u + \zeta v)\, d\zeta.$$

The Gonzales discrete gradient [19] is defined by

$$\bar{\nabla}H(u, v) = \nabla H\left(\frac{u + v}{2}\right) + \frac{H(v) - H(u) - \left\langle \nabla H\left(\frac{u+v}{2}\right), v - u\right\rangle}{\langle v - u, v - u\rangle}(v - u).$$

Another example is the Itoh–Abe discrete gradient [29]

$$\bar{\nabla}_{\text{ItohAbe}}H(u, v) := \begin{pmatrix} \frac{H(v_1, u_2, ..., u_d) - H(u_1, u_2, ..., u_d)}{v_1 - u_1} \\ \frac{H(v_1, v_2, ..., u_d) - H(v_1, u_2, ..., u_d)}{v_2 - u_2} \\ \vdots \\ \frac{H(v_1, ..., v_{d-1}, v_d) - H(v_1, ..., v_{d-1}, u_d)}{v_d - u_d} \end{pmatrix}.$$

## 4.2 Universal Energy-Behavior-Preserving Integrators

The above three discrete gradients are just examples from the family of maps $\bar{\nabla}H$ satisfying (5) and (6) above. *In this paper, we fully characterize the set of all possible discrete gradients as follows.*

Let $\bar{\nabla}_0 H$ be a particular choice of discrete gradient, henceforth denoted the *reference gradient*. Then, from (5) it follows that any other discrete gradient $\bar{\nabla}H(u, v)$ satisfies

$$\langle \bar{\nabla}H(u, v) - \bar{\nabla}_0 H(u, v), v - u\rangle = 0 \tag{7}$$

Thus, *the set of all discrete gradients can be described, and hence, parameterized for learning* as follows.

**Theorem 4.3** (Characterization of discrete gradients). *For a fixed function $H$, the set of all discrete gradient operators $\bar{\nabla}H : \mathbb{R}^d \times \mathbb{R}^d \to \mathbb{R}^d$ has the following parametrization*

$$\bar{\nabla}H(u, v) = \bar{\nabla}_0 H(u, v) + A(u, v)(v - u), \tag{8}$$

*where $A(u, v)$ is continuous and skew symmetric for all $u, v \in \mathbb{R}^d$, and where $\bar{\nabla}_0 H(u, v)$ is a fixed reference gradient satisfying (5) and (6).*

*Proof.* For the proof, see Appendix A. □

In fact, we can prove a stronger statement: *all energy-conserving numerical methods for the Hamilton equations can be written in this form*.

**Theorem 4.4.** *For a fixed function $H$, any energy-preserving integrators for the Hamilton equation*

$$\frac{\mathrm{d}u}{\mathrm{d}t} = S\nabla H(u)$$

*can be written as a discrete gradient scheme with a discrete gradient of the following form:*

$$\bar{\nabla} H(u,v) = \bar{\nabla}_0 H(u,v) + A(u,v)(v-u),$$

*where $A(u,v)$ is skew symmetric for all $u,v \in \mathbb{R}^d$ and $\bar{\nabla}_0 H(u,v)$ is a fixed reference gradient satisfying (5) and (6).*

*Proof.* See Appendix A for the proof. □

Note that we cannot obtain a similar theorem for dissipative systems in a straightforward way because, in the proof of the above theorem, we use the energy-conservation law of the conservative systems and also the non-degeneracy of $S$.

**Proposed Method** Although the discrete gradient method preserves the energy behaviors, it still introduces numerical errors that accumulate and grow over time. In this study, we propose a method for learning highly accurate energy-behavior-preserving integrators. The method can be summarized as Figure 1. This paper aims to optimize the discrete gradient based on the above characterization of discrete gradients, with the aim of not only preserving the energy but also yielding numerical solutions closer to the true solution.

The proposed method is formulated as follows:

$$\bar{\nabla}_{\mathrm{NN}} H(u^{(n)}, u^{(n+1)}) := \bar{\nabla} H(u^{(n)}, u^{(n+1)})$$
$$+ M_{\mathrm{NN}}(u^{(n)}, u^{(n+1)})(u^{(n+1)} - u^{(n)}), \quad (9)$$

where $M_{\mathrm{NN}}(u^{(n)}, u^{(n+1)})$ is a skew-symmetric matrix learned by a neural network that takes $u^{(n)}$ and $u^{(n+1)}$ as

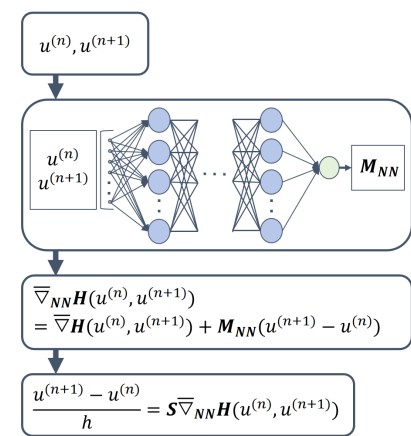

Figure 1: Outline of proposed method

the input. The universal approximation properties are the direct consequences of the above theorems.

**Theorem 4.5** (Universal approximation theorem for energy-preserving integrators for Hamiltonian systems)**.** *Suppose that the phase space is compact and $M_{\mathrm{NN}}$ is modeled by using a multilayer perceptron (MLP) with a sufficiently smooth activation function so that the MLPs admit the universal approximation property for the continuous skew symmetric matrices $M(u,v)$'s. Then the proposed method admits the universal approximation property for energy-preserving integrators for Hamiltonian systems.*

**Theorem 4.6** (Universal approximation theorem for energy-dissipative discrete-gradient integrators for dissipative systems)**.** *Suppose that the phase space is compact and $M_{\mathrm{NN}}$ is modeled by using a MLP with a sufficiently smooth activation function so that the MLPs admit the universal approximation property for the continuous skew symmetric matrices $M(u,v)$'s. Then the proposed method admits the universal approximation property for energy-dissipative discrete-gradient integrators for dissipative systems.*

Given a time series data $u^{(n)}$'s at time $t^{(n)}$'s, the proposed model is trained to minimize the squared error between the left- and right-hand sides of (9):

$$\text{minimize} \sum_n \left\| \frac{u^{(n+1)} - u^{(n)}}{h^{(n)}} - \bar{S}\bar{\nabla}_{\mathrm{NN}} H(u^{(n+1)}, u^{(n)}) \right\|_2^2, \quad (10)$$

where $h^{(n)} = t^{(n+1)} - t^{(n)}$.

If the matrix $M_{\mathrm{NN}}$ satisfies $M_{\mathrm{NN}}(u,v) = M_{\mathrm{NN}}(v,u)$, then the proposed integrator (9) at least has the second-order accuracy. Otherwise, the proposed integrator (9) is the first-order method [24].

# 5    Experiment

We used standard multilayer perceptrons with the hyperbolic tangent (tanh) as the activation function to learn the matrix $M_{\mathrm{NN}}$. The hyperparameters for training are provided in Appendix D.1. The optimizer is set to Adam [33]. As for the loss function, we chose MSELoss (Mean Squared Error Loss). Final loss values are provided in Appendix D.2. All experiments were performed on an Intel i5-13500H CPU, and computations were done in double precision.

In our experiments, we employ the AVF and Itoh–Abe methods as reference discrete gradient. As demonstrated in the experiments provided in the Appendix D.2, for the choice of the reference discrete gradient, we prefer to use the AVF method for its accuracy when it is computationally feasible. Otherwise, the Itoh-Abe method offers a simpler approach, making it more suitable for general equations. Moreover, to improve the performance of the Itoh–Abe method, we employ its time-symmetric version as the reference method in all subsequent experiments as follows:

$$\bar{\nabla}_{\mathrm{ItohAbe}_{\mathrm{SYM}}} H(u,v) = \left( \bar{\nabla}_{\mathrm{ItohAbe}} H(u,v) + \bar{\nabla}_{\mathrm{ItohAbe}} H(v,u) \right)/2.$$

Although the proposed method can theoretically guarantee at most second-order accuracy, we conduct comparisons with reference numerical methods using half step size and double steps, as well as with fourth-order AVF and Itoh–Abe methods constructed by the composition method [24].

The implementation code is available in supplementary material on OpenReview: `https://openreview.net/forum?id=G2uILEbcLF`.

## 5.1   AVF Discrete-Gradient-Based Method for Energy-Conservative Nonlinear System

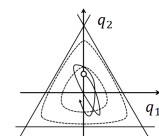

Figure 2:    Hénon-Heiles

**Hénon-Heiles System**    The Hénon-Heiles model (Figure 2) describes the stellar motion restricted to a plane. It shows chaotic motion and has a separable and polynomial Hamiltonian given by:

$$H(p,q) = \frac{1}{2}(p_1^2 + p_2^2) + \frac{1}{2}(q_1^2 + q_2^2) + q_1^2 q_2 - \frac{1}{3}q_2^3.$$

Since this is a polynomial Hamiltonian equation, the AVF method can be easily applied. Due to the chaotic characteristics of the system, even small errors can evolve into large deviations over long iterations. Although the AVF method conserves energy, it can still lead to significant errors for this system.

We use two sets of data: the flow data and the irregular data for the Hénon-Heiles equation. The training flow data $\{(u^{(n-1)}, u^{(n)}) \mid n = 1, ..., N\}$ consists of $N = 100$ pairs of points obtained from the trajectory calculated using a high-order integrator sampled with a time step $h = 0.3$, starting from $u^{(0)} = (q_1^{(0)}, q_2^{(0)}, p_1^{(0)}, p_2^{(0)}) = (0.3, 0.3, -0.2, -0.2)$. After training, we use the trained integrator to compute the flow starting at $u^{(N)}$ for 300 steps.

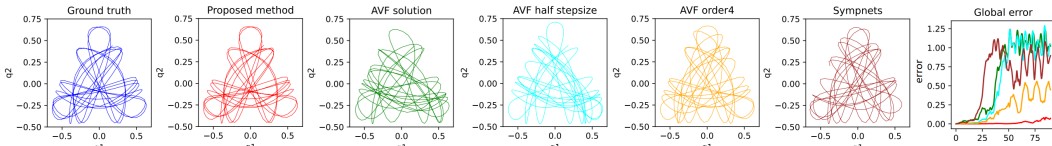

Figure 3: Results for the Hénon-Heiles eq. (predicted trajectory and global error.) The error plot uses the same color as the trajectory plots.

We also consider the computational time because the proposed numerical scheme is computationally more expensive than the AVF method due to the use of the neural network. When using a 10-layer, 100-width network for the Hénon-Heiles experiment, the computation time for the trajectory of 300 points falls between the AVF method and the AVF (half step size), which uses half the time step size and doubles the number of steps. For example, The computational times were: Proposed: 5.522 [sec], AVF: 3.608 [sec], AVF (half step size): 6.616 [sec]. In fact, the proposed method outperforms the fourth-order method, which offer higher accuracy while require more computation time than AVF(half step size). We also compared the trajectory computed using a 10-layer, 100-width G-sympnets trained for $100,000$ iterations with learning rate $0.001$ and the same training dataset by modifying the officially released code [31]. The predicted trajectory and the corresponding global

error ($\|u^{(n)} - u(nh)\|_2$) comparison are illustrated in Figure 3, from which we can deduce that the proposed method yields trajectory closer to the ground truth, and for details see Appendix D.4.

We further conducted experiments using irregular data. The training dataset $\{((u^{(i-1)}, h^{(i)}), u^{(i)}) \mid i = 1, \ldots, I\}$ consists of $I = 100$ one-step input-output pairs, where the elements of $u^{(i-1)}$ are randomly sampled from the domain $[-1, 1]$, and the corresponding step sizes $h^{(i)}$ are randomly chosen from $[0, 0.5]$. Each $u^{(i)}$ represents the one-step solution computed from $u^{(i-1)}$ using step size $h^{(i)}$. After training, we simulate the flow starting from the initial state $u^{(0)} = (q_1^{(0)}, q_2^{(0)}, p_1^{(0)}, p_2^{(0)}) = (0.3, 0.3, -0.2, -0.2)$ using a fixed step size $h = 0.3$ for 100 time steps. The numerical results indicate that the proposed method yields significantly improved accuracy over baseline methods. For detailed comparisons, please refer to Appendix D.4.

We also conducted experiments on the pendulum system to validate the generality of our proposed method. Detailed experimental settings and results are provided in Appendix D.3.

## 5.2 Itoh-Abe Discrete-Gradient-Based Method for Energy-Conservative Nonlinear System

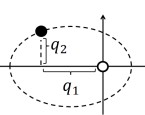

**Two-body Problem** In the two-body problem, the motion of two bodies that attract each other is considered. We assume that one of the bodies is fixed at the center of the coordinate system and the position of the other one is given as $q = (q_1, q_2)$ (Figure 4.) The Hamiltonian is given as follows:

Figure 4: Two-body

$$H(q_1, q_2, p_1, p_2) = \frac{1}{2}(p_1^2 + p_2^2) - \frac{1}{\sqrt{q_1^2 + q_2^2}}.$$

The training flow data $\{(u^{(n-1)}, u^{(n)}) \mid n = 1, \ldots, N\}$ consists of $N = 30$ pairs with a fixed time step $h = 0.1$, starting from $u^{(0)} = (q_1^{(0)}, q_2^{(0)}, p_1^{(0)}, p_2^{(0)}) = (-0.8, 0, 0, 1)$. After training, we use the trained integrator to compute the flow starting at $u^{(N)}$ for 300 steps.

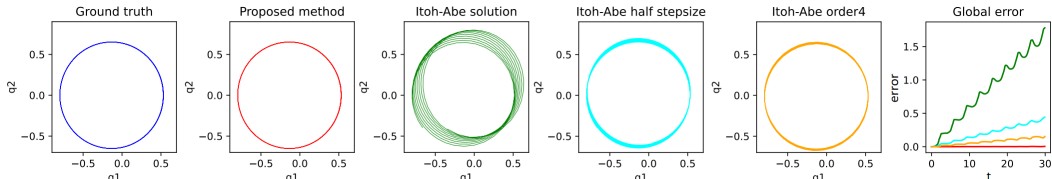

Figure 5: Results of the Two-body (predicted trajectory and global error.) The error plot uses the same color as the trajectory plots.

Figure 5 compares the trajectories predicted by the proposed method with Itoh-Abe methods. While the Itoh–Abe methods yield solutions that generally follow the elliptical shape of the ground-truth, they exhibit noticeable deviations from the true trajectory. For details, see Appendix D.4.

**Double Pendulum** The double pendulum is known to exhibit complex behaviors with a strong sensitivity to initial conditions, which makes the computation very challenging.

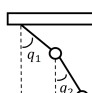

This system has a non-polynomial Hamiltonian as follows:

$$H(q_1, q_2, p_1, p_2) = \frac{p_1^2 + 2p_2^2 - 2p_1p_2\cos(q_1 - q_2)}{2(1 + \sin^2(q_1 - q_2))} - 2\cos q_1 - \cos q_2.$$

Figure 6: 2-pendulum

The training flow data $\{(u^{(n-1)}, u^{(n)}) \mid n = 1, \ldots, N\}$ consists of $N = 100$ pairs with a fixed time step $h = 0.3$, starting from $u^{(0)} = (q_1^{(0)}, q_2^{(0)}, p_1^{(0)}, p_2^{(0)}) = (\frac{1}{6}\pi, \frac{1}{8}\pi, 0, 0)$. After training, we use the trained integrator to compute the flow starting at $u^{(N)}$ for 300 steps.

Although the chaotic nature of the double pendulum makes learning difficult, the proposed method provides a solution that is close to the true trajectory (Figure 7). For more details, see Appendix D.4.

## 5.3 AVF Discrete-Gradient-Based Method for Dissipative Nonlinear System

The discrete gradient methods can also be applied to dissipative systems

$$\frac{\mathrm{d}}{\mathrm{d}t}\begin{pmatrix} q \\ p \end{pmatrix} = \begin{pmatrix} 0 & 1 \\ -1 & -\delta \end{pmatrix} \nabla H$$

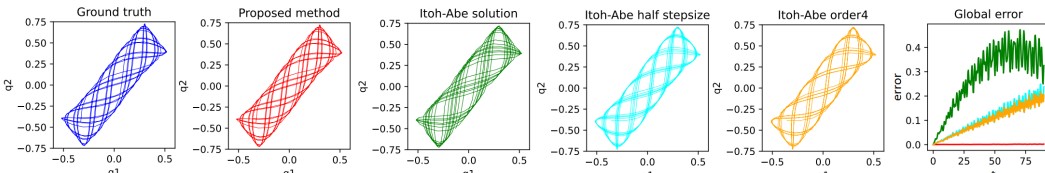

Figure 7: Results of the double pendulum (predicted trajectory and global error.) The error plot uses the same color as the trajectory plots.

to construct energy-dissipating integrators, where $\delta$ denotes the dissipative coefficient. Therefore, our proposed approach can be employed to optimize them.

**Dissipative Pendulum**    The Hamiltonian for the dissipative pendulum is the same as that for the pendulum $H(q,p) = \frac{1}{2}p^2 - \cos(q)$.

The dissipation coefficient was set to $\delta = 0.01$. We train the network with the flow data $\{(u^{(n-1)}, u^{(n)}) \mid n = 1, ..., N\}$ consists of $N = 100$ pairs with a fixed time step $h = 0.3$,

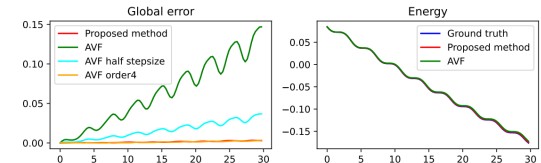

Figure 8: Results of the dissipative pendulum

starting from $u^{(0)} = (q^{(0)}, p^{(0)}) = (2, 0)$. After training, the trained integrator was used to compute the flow starting at $u^{(N)}$ for 100 steps. As seen in Figure 8, the accuracy of the integrator was greatly improved again by the proposed method while preserving the dissipation property.

**Duffing Oscillator**    The Duffing equation is a non-linear differential equation that models certain damped oscillators with the Hamiltonian:

$$H(q,p) = \frac{1}{2}p^2 + \frac{1}{2}\alpha q^2 + \frac{1}{4}\beta q^4$$

where we set the linear stiffness coefficient to $\alpha = 1$, the non-linearity coefficient to $\beta = -5$ and the damping coefficient to $\delta = 0.05$. We train the flow data $\{(u^{(n-1)}, u^{(n)}) \mid n = 1, ..., N\}$ consists of $N = 200$ pairs with a fixed time step $h = 0.1$, starting from $(q^{(0)}, p^{(0)}) = (4, 0)$. After training, we computed the flow starting at $u^{(N)}$ for 300 steps.

Figure 9 shows that the proposed method again improves the accuracy; however, compared to energy-conserving systems, the numerical error is slightly larger. This is because the discrete gradient method only guarantees the energy dissipation property but does not ensure the accuracy of energy for energy-dissipative systems.

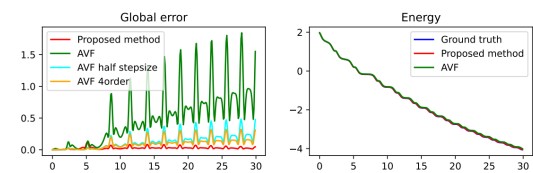

Figure 9: Results of the Duffing oscillator

### 5.4   AVF Discrete-Gradient-Based Method for Energy-Conservative Nonlinear PDEs

As the discrete gradient method is applicable to semi-discretized PDEs, our approach is also applicable. We applied the proposed method to a semi-discretized Hamiltonian PDE, the Korteweg-de Vries (KdV) equation:

$$\frac{\partial u}{\partial t} = -6u\frac{\partial u}{\partial x} - \frac{\partial^3 u}{\partial x^3}.$$

The semi-discrete Hamiltonian is defined as $H(u) = \sum_j \left( \frac{1}{2(\Delta x)^2}(u_{j+1} - u_j)^2 - u_j^3 \right)$, where the index $j$ denotes spatial discretization points at a fixed time level with the spatial step size $\Delta x$.

This equation is a partial differential equation that describes shallow water waves. We consider this equation on the interval $[0, 20]$ under the periodic boundary condition. We used the training data with the time length of $T = 30$ and the time step size of $\Delta t = 0.01$. The spatial step size was $\Delta x = 0.5$. For testing,

Figure 10: Results of KdV

the computation was carried out from the time length $T = 30$, with the starting being at the final state of the training dataset.

The numerical errors shown in Figure 10 confirm that the proposed method yields more accurate solutions compared to the AVF method. For more details, see Appendix D.4.

## 5.5 Supplementary Experiments

**Impact of Training Data**    We also conducted comparative experiments to investigate how varying the size of irregularly training data influences the performance of the learned integrator. The experimental results demonstrate that increasing the amount of training data is beneficial for improving the training performance of the learned integrator. See Appendix D.2 for details

**Comparison with Other Methods**    For completeness of exposition, we compare our approach with other methods. Our method focuses on learning structure-preserving numerical integrators, whereas many existing approaches (e.g., PINNs) aim to directly approximate continuous solutions. Because these objectives differ, such comparisons are inherently difficult and should be interpreted with care. We therefore report supplementary results with PINNs and TRS-ODEN on standard dynamical systems. The protocol and complete results are given in Appendix D.5. Overall, our method shows smaller errors and more stable long-horizon trajectories.

**Computational Time**    One limitation of the method is the increased computational time associated with neural networks during training and inference. In particular, the proposed method requires time for data preparation and network training. To see the actual increase in total computation time, we conducted additional numerical experiments. Specifically, we discuss the computation time in detail, especially for the case of the KdV equation, which is the largest system in the experiments. For other experiments, see Appendix D.6. Also, GPUs can be used for training to accelerate calculations. In the experiment, we used an Intel Xeon 6900P CPU and an AMD Instinct MI300A GPU for computation.

We first set the number of nodes for spatial discretization of the KdV equation to 100. The computation time for generating the training data was 28.709 sec using a CPU and the scipy odeint function. The training time was 6.407 sec using a MI300A GPU. The integration time during testing was 52.738 (proposed), 22.785 (2nd order AVF) and 127.280 (4th order AVF). Thus the total computational time of the proposed method was $28.709 + 6.407 + 52.738 = 87.854$, which is less than the 127.280 sec required by the 4th order AVF method. Similarly, if we set the number of the nodes to 120, the computational time for each method was 50.957 (data generation) + 8.311 (training) + 76.460 (integration) = 135.728 (proposed), 33.034 (2nd order AVF), 200.003 (4th order AVF). The total computational time of the proposed method was again shorter than that of the 4th-order AVF method. Because our proposed approach typically outperforms the 4th-order AVF method, the proposed approach can compute more accurate numerical solutions in a shorter amount of time, even when the training time and the time for generating data are included.

A closer look at the execution time of the above experiments shows that the computation of the neural network itself is actually not so large. Rather, the matrix-vector multiplication of the skew-symmetric matrix affects the computational complexity. However, the effect of this matrix multiplication does not seem to be so significant, since the multiplication with the skew-symmetric or negative-definite matrix placed in front of the gradient of the energy takes about the same amount of computation. It is also possible to reduce computational cost by low-rank approximation of the learned matrices. In addition, our method does not necessarily need to be used at all time steps. By using it only when higher computational accuracy is desired, computational time can be further reduced.

## 6 Conclusion

In this work, we proposed a method for learning numerical integrators that preserve the energy conservation or dissipation laws with universal approximation properties. We validated the proposed approach across a diverse set of systems, including chaotic nonlinear Hamiltonian systems, dissipative systems, and a nonlinear partial differential equation. In all scenarios, the method exhibited advantages, particularly in complex or chaotic regimes. With further optimization, we believe this method will play an important role in application areas in the future.

## Acknowledgments and Disclosure of Funding

Funding in direct support of this work: JST CREST Grant Number JPMJCR1914 and JPMJCR24Q5, JSPS KAKENHI Grant Number 25K15148, JST ASPIRE JPMJAP2329 and NIFS Collaborative Research NIFS25KISC015. This work is also supported by the Horizon Europe, MSCA-SE project 101131557 (REMODEL).

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

# A    Proofs

*Proof of Theorem 3.1.* By using the chain rule, it holds that

$$\frac{\mathrm{d}H}{\mathrm{d}t} = \nabla H^\top \frac{\mathrm{d}u}{\mathrm{d}t}.$$

Substitution of the equation yields

$$\frac{\mathrm{d}H}{\mathrm{d}t} = \nabla H^\top S \nabla H \begin{cases} = 0 & \text{(if } S \text{ is skew symmetric)} \\ \leq 0 & \text{(if } S \leq O.) \end{cases}$$

□

*Proof of Theorem 4.2.* The proof is an discrete analogue of the proof of Theorem 3.1. By using the discrete chain rule, it holds that

$$\frac{H(u^{(n+1)}) - H(u^{(n)})}{h} = \bar{\nabla} H^\top \frac{u^{(n+1)} - u^{(n)}}{h}.$$

Substitution of the discrete gradient scheme yields

$$\frac{H(u^{(n+1)}) - H(u^{(n)})}{h} = \bar{\nabla} H^\top \bar{S} \bar{\nabla} H \begin{cases} = 0 & \text{(if } \bar{S} \text{ is skew symmetric)} \\ \leq 0 & \text{(if } \bar{S} \leq O.) \end{cases}$$

□

*Proof of Theorem 4.3.* First, note that if a vector $w$ is perpendicular to another vector $v - u$, then there exists a skew symmetric matrix $A$ such that $w = A(v - u)$. Hence, the proof is completed if we show that all discrete gradient operators have the following representation:

$$\bar{\nabla} H(u, v) = \bar{\nabla}_0 H(u, v) + w(u, v), \text{ where } w(u, v) \perp v - u \text{ for all } u, v \in \mathbb{R}^d.$$

Let $\bar{\nabla} H(u, v)$ be of the form (8) for some $w(u, v)$, and we check that it satisfies (5) and (6).

$$\langle \bar{\nabla} H(u, v), v - u \rangle = \langle \bar{\nabla}_0 H(u, v), v - u \rangle + \langle w(u, v), v - u \rangle = \langle \bar{\nabla}_0 H(u, v), v - u \rangle = H(v) - H(u).$$

By definition, $\bar{\nabla} H(u, u) = \bar{\nabla}_0 H(u, u) + w(u, u) = \nabla H(u)$. Conversely, if $\bar{\nabla} H(u, v)$ is a discrete gradient, then define $w(u, v) = \bar{\nabla} H(u, v) - \bar{\nabla}_0 H(u, v)$ which is continuous. By (7), $w(u, v) \perp v - u$, and $w(u, u) = \bar{\nabla} H(u, u) - \bar{\nabla}_0 H(u, u) = 0$ by (6) since both of the discrete gradients are consistent. □

*Proof of Theorem 4.4.* Suppose that a numerical scheme

$$\frac{u^{(n+1)} - u^{(n)}}{h} = f(u^{(n+1)}, u^{(n)}) \tag{11}$$

preserves the energy

$$H(u^{(n+1)}) = H(u^{(n)}).$$

Let $u^{(n+1)}$ and $u^{(n)}$ be solutions to (11). It is sufficient to show that $f$ can be written as

$$f(u, v) = S\left(\bar{\nabla}_0 H(u, v) + w(u, v)\right), \text{ where } w(u, v) \perp v - u$$

with a reference discrete gradient $\bar{\nabla}_0 H(u, v)$ of $H$.

From the property of the discrete gradient and the energy-conservation property, we get

$$0 = \frac{H(u^{(n+1)}) - H(u^{(n)})}{h} = \bar{\nabla}_0 H(u^{(n+1)}, u^{(n)})^\top \frac{u^{(n+1)} - u^{(n)}}{h} = \bar{\nabla}_0 H(u^{(n+1)}, u^{(n)})^\top f(u^{(n+1)}, u^{(n)}).$$

Hence $f$ must be perpendicular to the discrete gradient $\bar{\nabla}_0 H$. Because we assume that the target system is a Hamiltonian equation, the matrix $S$ is invertible. Therefore, we have

$$(S\bar{\nabla}_0 H(u^{(n+1)}, u^{(n)}))^\top S^{-1} f(u^{(n+1)}, u^{(n)}) = \bar{\nabla}_0 H(u^{(n+1)}, u^{(n)})^\top S^\top S^{-1} f(u^{(n+1)}, u^{(n)})$$

$$= -\bar{\nabla}_0 H(u^{(n+1)}, u^{(n)})^\top f(u^{(n+1)}, u^{(n)}) = 0$$

Thus, $S^{-1}f$ is in the orthogonal complement $\mathcal{M}$ of $S\bar{\nabla}_0 H$. Because $S$ is skew-symmetric, $\bar{\nabla}_0 H$ is perpendicular to $S\bar{\nabla}_0 H$, which means that $\bar{\nabla}_0 H$ is in the orthogonal complement $\mathcal{M}$. Therefore, we can obtain an orthogonal basis $\{e_1, e_2, \ldots, e_N\}$ of $\mathcal{M}$, where $e_1 = \bar{\nabla}_0 H$ and $N = \dim \mathcal{M}$. By using this basis, $S^{-1}f$ can be written as

$$S^{-1}f = c_1 \bar{\nabla}_0 H + e$$

where $c_1 \in \mathbb{R}$ and $e \in \text{span}\{e_2, ..., e_N\}$. We want to show that $e$ is perpendicular to $u^{(n+1)} - u^{(n)} = hf(u^{(n+1)}, u^{(n)})$. The inner product of $e$ and $f$ is

$$\langle u^{(n+1)} - u^{(n)}, e \rangle = h\langle f, e \rangle = h\langle f, S^{-1}f - c_1 h \bar{\nabla}_0 H \rangle = h\langle f, S^{-1}f \rangle - c_1 h\langle f, \bar{\nabla}_0 H \rangle = 0$$

because $S^{-1}$ is skew-symmetric, $\langle f, f \rangle = 0$. Hence $\langle f, \bar{\nabla}_0 H \rangle = 0$. Thus $e$ is perpendicular to $u^{(n+1)} - u^{(n)}$, so there exists a skew-symmetric matrix $A(u^{(n+1)}, u^{(n)})$ such that $e = A(u^{(n+1)}, u^{(n)})(u^{(n+1)} - u^{(n)})$.

Finally, we show that $c_1 = 1$. To this end, we take the limit $h \to 0$ in

$$S^{-1}f = c_1 \bar{\nabla}_0 H + e = c_1 \bar{\nabla}_0 H + A(u^{(n+1)}, u^{(n)})(u^{(n+1)} - u^{(n)}).$$

Since the scheme is consistent, $S^{-1}f \to \nabla H$ as $h \to 0$. On the other hand, the right-hand side tends to $c_1 \nabla H$ because of the consistency condition (6) of discrete gradients. Hence we get $\nabla H = c_1 \nabla H$ and hence $c_1 = 1$. $\qquad\square$

## B  Geometric Description of Target Differential Equations

The target equations in this study are characterized as gradient flows on symplectic or Riemannian manifolds. Let $(\mathcal{M}, \omega)$ be a symplectic manifold and $(\mathcal{M}, g)$ be a Riemannian manifold. For symplectic manifolds, $\omega$ is a symplectic 2-form. In Riemannian manifolds, $g$ denotes the inner product. These define a bilinear function $\omega_u : T_u\mathcal{M} \times T_u\mathcal{M} \to \mathbb{R}$ or $g_u : T_u\mathcal{M} \times T_u\mathcal{M} \to \mathbb{R}$ for each $u \in \mathcal{M}$. It is assumed that $\omega$ and $g$ are nondegenerate in the sense that the matrix $M_u$ representing these bilinear functions is nondegenerate for any $u \in \mathcal{M}$.

For a given energy function $H : \mathcal{M} \to \mathbb{R}$, we can define differential equations

$$\dot{u} = X, \quad \omega(X, \cdot) = \mathrm{d}H(\cdot)$$

on symplectic manifolds and

$$\dot{u} = X, \quad g(X, \cdot) = -\mathrm{d}H(\cdot)$$

on Riemannian manifolds, where $\mathrm{d}H : T\mathcal{M} \to \mathbb{R}$ is the differential of $H$. For symplectic manifolds, this equation is the Hamilton equation, and for Riemannian manifolds, it is the gradient flow.

These equations have energy conservation and the energy dissipation laws, respectively. In fact, in Hamiltonian systems, it holds that

$$\frac{\mathrm{d}H}{\mathrm{d}t} = \mathrm{d}H(X) = \omega(X, X) = 0$$

because $\omega_u$ is skew-symmetric due to the property of symplectic forms. Also, for Riemannian manifolds,

$$\frac{\mathrm{d}H}{\mathrm{d}t} = \mathrm{d}H(X) = -g(X, X) \leq 0$$

By using the relation between the gradient and the derivative

$$\mathrm{d}H(X) = \langle \nabla H, X \rangle$$

these equations can be rewritten as vector-matrix representations:

$$\frac{\mathrm{d}u}{\mathrm{d}t} = M_u \nabla H$$

for symplectic manifolds and

$$\frac{\mathrm{d}u}{\mathrm{d}t} = -M_u \nabla H$$

for Riemannian manifolds, respectively. These equations are the target equations of this study.

## C  Target Partial Differential Equations and Its Semi-Discretization

The proposed method in this paper can also be applied to semi-discretized partial differential equations in the spatial direction. Typical examples of the target partial differential equations are equations of the following form

$$\frac{\partial u}{\partial t} = \left(\frac{\partial}{\partial x}\right)^{\alpha} \frac{\delta H}{\delta u}, \tag{12}$$

where $H$ is the energy density, which is a function of $u$, $u_x$, $u_{xx}$, and so on. $u_x$ denotes $\frac{\partial u}{\partial x}$, and $u_{xx}$ denotes $\frac{\partial^2 u}{\partial x^2}$. $\delta H/\delta u$ is called the variational derivative of $H$. For example, if $H$ is a function of $u$ and $u_x$, then this is defined by

$$\frac{\delta H}{\delta u} = \frac{\partial H}{\partial u} - \frac{\partial}{\partial x}\frac{\partial H}{\partial u_x}.$$

Equations of the form (12) include many partial differential equations, such as the advection equation, the KdV equation, the Burgers equation, the Allen–Cahn equation, and the Cahn–Hilliard equation. Equations that are slightly different but can be expressed in a similar form include the Maxwell equation, the wave equation, the Klein–Gordon equation, and the nonlinear Schrodinger equation. Although the above equations are defined in one-dimensional space, they can be extended to multiple dimensions in a straightforward way. See, e.g., Furihata and Matsuo [18] for details.

It is known that (12) has the global energy conservation law or the energy dissipation or increasing law under certain boundary conditions, such as the periodic boundary condition. Suppose that the equation is defined on the interval $[0, L]$. For simplicity, we assume the periodic boundary condition. In this case, the time differentiation of the total energy yields

$$\frac{\mathrm{d}}{\mathrm{d}t}\int_0^L H\mathrm{d}x = \int_0^L \left(\frac{\partial H}{\partial u}u_t + \frac{\partial H}{\partial u_x}u_{xt}\right)\mathrm{d}x = \int_0^L \left(\frac{\partial H}{\partial u} - \frac{\partial}{\partial x}\frac{\partial H}{\partial u_x}\right)u_t\mathrm{d}x + \left[\frac{\partial H}{\partial u_x}u_t\right]_0^L.$$

Because we assume the periodic boundary condition, the boundary term must vanish. Therefore, we have

$$\frac{\mathrm{d}}{\mathrm{d}t}\int_0^L H\mathrm{d}x = \int_0^L \left(\frac{\partial H}{\partial u} - \frac{\partial}{\partial x}\frac{\partial H}{\partial u_x}\right)u_t\mathrm{d}x.$$

Substituting the definition of the variational derivatives and also the equation into the above equality, we obtain

$$\frac{\mathrm{d}}{\mathrm{d}t}\int_0^L H\mathrm{d}x = \int_0^L \left(\frac{\partial H}{\partial u} - \frac{\partial}{\partial x}\frac{\partial H}{\partial u_x}\right)u_t\mathrm{d}x = \int_0^L \frac{\delta H}{\delta u}\left(\frac{\partial}{\partial x}\right)^{\alpha}\frac{\delta H}{\delta u}\mathrm{d}x.$$

If $\alpha$ is odd and expressed as $\alpha = 2n + 1$ with an integer $n$, then by repeating the integration by parts, we get

$$\int_0^L \frac{\delta H}{\delta u}\left(\frac{\partial}{\partial x}\right)^{2n+1}\frac{\delta H}{\delta u}\mathrm{d}x = (-1)^n \int_0^L \left(\frac{\partial}{\partial x}\right)^n \frac{\delta H}{\delta u}\left(\frac{\partial}{\partial x}\right)^{n+1}\frac{\delta H}{\delta u}\mathrm{d}x$$

$$= (-1)^{n+1}\int_0^L \left(\frac{\partial}{\partial x}\right)^{n+1}\frac{\delta H}{\delta u}\left(\frac{\partial}{\partial x}\right)^n\frac{\delta H}{\delta u}\mathrm{d}x.$$

Therefore, we have

$$(-1)^n\int_0^L \left(\frac{\partial}{\partial x}\right)^n\frac{\delta H}{\delta u}\left(\frac{\partial}{\partial x}\right)^{n+1}\frac{\delta H}{\delta u}\mathrm{d}x = -(-1)^n\int_0^L \left(\frac{\partial}{\partial x}\right)^{n+1}\frac{\delta H}{\delta u}\left(\frac{\partial}{\partial x}\right)^n\frac{\delta H}{\delta u}\mathrm{d}x$$

and hence the energy conservation law follows:

$$2(-1)^n\int_0^L \left(\frac{\partial}{\partial x}\right)^n\frac{\delta H}{\delta u}\left(\frac{\partial}{\partial x}\right)^{n+1}\frac{\delta H}{\delta u}u_t\mathrm{d}x = 0.$$

Similarly, when $\alpha = 2n$, we have

$$\int_0^L \frac{\delta H}{\delta u}\left(\frac{\partial}{\partial x}\right)^{2n+1}\frac{\delta H}{\delta u}u_t\mathrm{d}x = (-1)^n\int_0^L \left[\left(\frac{\partial}{\partial x}\right)^n\frac{\delta H}{\delta u}\right]^2 u_t\mathrm{d}x.$$

Thus, the energy decreasing or increasing law follows. Thus, it can be seen that these equations have the same energy behavior as the target ordinary differential equations of this study.

Next, we semi-discretize these partial differential equations only in the spatial direction using the finite difference method while preserving the energy behaviors. We devide the interval $[0, L]$ into $M$ equal parts with the spatial step size $\Delta x$. The approximation of $u(t, j\Delta x)$ is denoted by $u_j(t)$, and we donote the vector $(u_0(t), ..., u_M(t))$ as $\vec{u}$. Let $D_1$, $D_+$, and $D_-$ be difference matrices representing the central, forward, and backward differences, respectively:

$$D_1 = \frac{1}{2\Delta x} \begin{pmatrix} 0 & 1 & & & -1 \\ -1 & 0 & 1 & & \\ & & \ddots & & \\ & & -1 & 0 & 1 \\ 1 & & & -1 & 0 \end{pmatrix}, \ D_+ = \frac{1}{2\Delta x} \begin{pmatrix} -1 & 1 & & \\ & -1 & 1 & \\ & & \ddots & \\ & & -1 & 1 \\ 1 & & & -1 \end{pmatrix}, \ D_- = \frac{1}{2\Delta x} \begin{pmatrix} 1 & & & -1 \\ -1 & 1 & & \\ & \ddots & & \\ & -1 & 1 & \\ & & -1 & 1 \end{pmatrix}.$$

First, the energy function can be approximated by approximating $u_x$ in the energy function by using the finite difference

$$\int_0^L H(u, u_x)\mathrm{d}x \simeq H_\mathrm{d}(\vec{u}, D_1\vec{u}, D_+\vec{u}, D_-\vec{u}) =: \hat{H}_\mathrm{d}(\vec{u})$$

We also discretize the differential operator

$$\left(\frac{\partial}{\partial x}\right)^\alpha \tag{13}$$

in the equation. For $\alpha = 2n$, we can use $n$ copies of $D_+$ and $D_-$ to form a matrix

$$D_+ \cdots D_+ D_- \cdots D_-$$

to approximate (13). This matrix is positive or negative semi-definite, since $D_+^\top = -D_-$. Thus, the semi-discretized equation

$$\frac{\mathrm{d}\vec{u}}{\mathrm{d}t} = D_+ \cdots D_+ D_- \cdots D_- \nabla\hat{H}_\mathrm{d}(\vec{u})$$

is an ordinary differential equation with the energy dissipating or increasing property. Similarly, if $\alpha$ is $2n + 1$, then the equation is semi-discretized into an ordinary differential equation with an energy conservation law:

$$\frac{\mathrm{d}\vec{u}}{\mathrm{d}t} = D_+ \cdots D_+ D_1 D_- \cdots D_- \nabla\hat{H}_\mathrm{d}(\vec{u}).$$

Note that $D_+ \cdots D_+ D_1 D_- \cdots D_-$ is skew-symmetric because $D_1$ is skew-symmetric. The proposed method yields numerical integrators that preserve the energy behavior for each of these semi-discretized equations.

## D  Supplementary experiments

### D.1  hyperparameters

Table 2 shows the hyperparameters for training.

### D.2  Experiment loss

**Loss under Different Reference Methods with Varying Network Depths and Widths**  We use the Hénon-Heiles model:

$$H(p, q) = \frac{1}{2}(p_1^2 + p_2^2) + \frac{1}{2}(q_1^2 + q_2^2) + q_1^2 q_2 - \frac{1}{3}q_2^3.$$

The training flow data $\{(u^{(n-1)}, u^{(n)}) \mid n = 1, ..., N\}$ consists of $N = 100$ pairs of points obtained from the trajectory calculated using a high-order integrator sampled with a time step $h = 0.3$, starting from $u^{(0)} = (q_1^{(0)}, q_2^{(0)}, p_1^{(0)}, p_2^{(0)}) = (0.3, 0.3, -0.2, -0.2)$. After training, we use the flow starting at $u^{(N)}$ for 300 steps as test.

Table 2: hyperparameters

| PROBLEM | LR | EPOCHS | LAYER | WIDTH |
|---|---|---|---|---|
| PENDULUM | | | | |
| -FLOW | 0.0001 | 10000 | 5 | 50 |
| -IRREGULAR | 0.0001 | 10000 | 5 | 50 |
| -DISSPATIVE | 0.0001 | 10000 | 5 | 50 |
| HÉNON-HEILES | | | | |
| -FLOW | 0.00001 | 30000 | 10 | 100 |
| -IRREGULAR | 0.00001 | 30000 | 10 | 100 |
| 2-BODY | 0.0001 | 10000 | 5 | 50 |
| 2-PENDULUM | 0.0001 | 10000 | 5 | 50 |
| DUFFING | 0.0001 | 10000 | 5 | 50 |
| KDV | 0.0001 | 10000 | 5 | 200 |

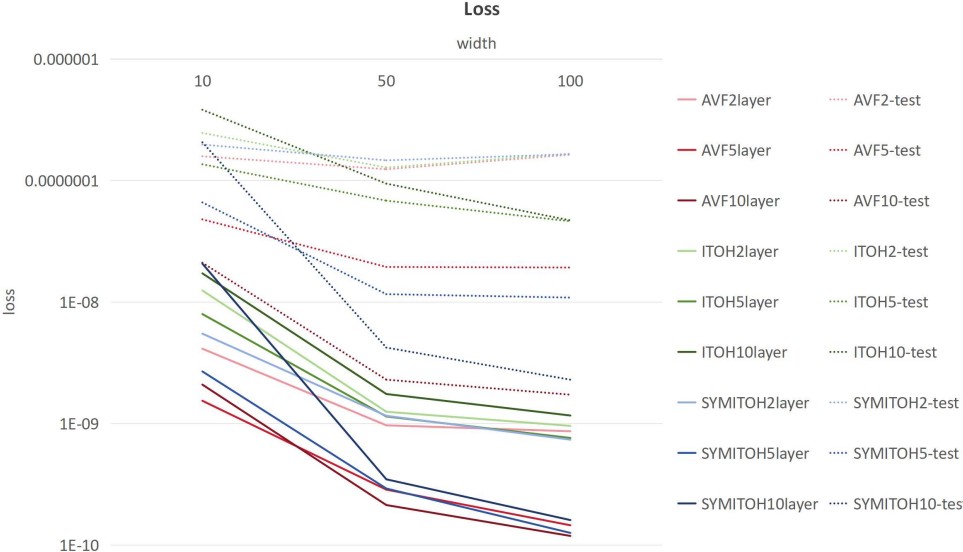

Figure 11: Loss for Different Reference Methods with Varying Network Depths and Widths

We conducted experiments using both AVF, Itoh-Abe method and time-symmetric Itoh-Abe method as baseline schemes. The learning rate was set to 0.00001, and the number of epochs was 30,000. The network architecture was varied with 2, 5, and 10 layers, and width of 10, 50, and 100 per layer. For each configuration, we performed five independent training runs and reported the final loss as results, as shown in the Figure 11.

As expected, the AVF-based method, being a second-order scheme, generally outperformed the Itoh-Abe method.

Based on the above experimental results and observations, we conclude that in the initial selection of the baseline method, the AVF scheme should be prioritized whenever it is computationally feasible, due to its higher accuracy and time symmetry. In cases where the original Hamiltonian system is too complex for the AVF discrete gradient to be computed, the Itoh–Abe method can serve as a practical alternative. Furthermore, it is worth exploring the possibility of constructing a time-symmetric version of the Itoh–Abe method to enhance its accuracy, potentially achieving a second-order scheme.

For this reason, we adopt a time-symmetric construction of Itoh–Abe method to achieve second-order. And the results in the Figure 11 indicate that the time-symmetric Itoh–Abe method outperforms the standard one.

$$\bar{\nabla}_{\text{ItohAbe}_{\text{SYM}}} H(u, v) = (\bar{\nabla}_{\text{ItohAbe}} H(u, v) + \bar{\nabla}_{\text{ItohAbe}} H(v, u))/2$$

**Experiment loss**   The training loss was computed by the mean-squared-error between $\frac{u_{\text{train}}^{(n)} - u_{\text{train}}^{(n-1)}}{h_{\text{train}}^{(n)}}$ and $\bar{S}\bar{\nabla}_{\text{NN}}H(u_{\text{train}}^{(n)}, u_{\text{train}}^{(n-1)})$, and the test loss was computed by the mean-squared-error between $\frac{u_{\text{test}}^{(n)} - u_{\text{test}}^{(n-1)}}{h_{\text{test}}^{(n)}}$ and $\bar{S}\bar{\nabla}_{\text{NN}}H(u_{\text{test}}^{(n)}, u_{\text{test}}^{(n-1)})$. Table 3 shows the losses in the form of mean $\pm$ standard deviation measured from 5 independent experiments. Note that all computations were performed in double precision. It can be seen that the models were trained very accurately.

Table 3: The values of the train and test loss.

| PROBLEM | TRAIN LOSS | TEST LOSS |
|---|---|---|
| PENDULUM | | |
| -FLOW | $7.75 \times 10^{-10} \pm 7.46 \times 10^{-10}$ | $1.95 \times 10^{-8} \pm 2.21 \times 10^{-8}$ |
| -IRREGULAR | $1.33 \times 10^{-7} \pm 1.07 \times 10^{-7}$ | $6.08 \times 10^{-7} \pm 3.95 \times 10^{-7}$ |
| -DISSPATIVE | $4.69 \times 10^{-9} \pm 2.06 \times 10^{-9}$ | $1.70 \times 10^{-8} \pm 9.67 \times 10^{-9}$ |
| HÉNON-HEILES | | |
| -FLOW | $1.19 \times 10^{-10} \pm 2.64 \times 10^{-11}$ | $1.72 \times 10^{-9} \pm 4.64 \times 10^{-10}$ |
| -IRREGULAR | $1.40 \times 10^{-8} \pm 2.71 \times 10^{-9}$ | $2.14 \times 10^{-8} \pm 5.22 \times 10^{-9}$ |
| 2-BODY | $1.14 \times 10^{-8} \pm 3.21 \times 10^{-9}$ | $2.92 \times 10^{-8} \pm 1.01 \times 10^{-8}$ |
| 2-PENDULUM | $6.34 \times 10^{-8} \pm 4.06 \times 10^{-8}$ | $8.69 \times 10^{-8} \pm 5.79 \times 10^{-8}$ |
| DUFFING | $9.24 \times 10^{-7} \pm 3.95 \times 10^{-7}$ | $7.73 \times 10^{-6} \pm 5.91 \times 10^{-6}$ |
| KDV | $1.09 \times 10^{-6} \pm 1.25 \times 10^{-10}$ | $1.69 \times 10^{-6} \pm 1.61 \times 10^{-10}$ |

**Experiment loss with different data and initial conditions**   We conducted additional experiments on both the Hénon-Heiles and the double pendulum. The aim was to evaluate the impact of training data distribution, data size, and initial conditions on the performance of the learned integrator.

*Hénon-Heiles system:* We used irregular training data $\{((u^{(i-1)}, h^{(i)}), u^{(i)}) \mid i = 1, ..., I\}$, where the elements of $u^{(i-1)}$ are randomly sampled from the domain $[-1, 1]$ and each step size $h^{(i)}$ is drawn independently from $[0, 0.5]$. Each $u^{(i)}$ represents the one-step solution computed from $u^{(i-1)}$ using step size $h^{(i)}$ with a high-order solver.

We performed experiments using $100, 1000, 5000$ training samples, respectively. The neural network consists of 10 layers, each with 100 width. We set the learning rate to $0.00001$ and trained the model for 30000 iterations. For testing, we selected random initial conditions from $[-0.4, 0.4]^4$. For each test trajectory, a fixed step size $h$ was randomly chosen from $[0, 0.5]$ and applied over 100 time steps.

*Double pendulum:* We generated training data as follows. First, we randomly sampled initial angles $(q1, q2)$ from the interval $[-0.8, 0.8]$, and set the initial momenta $(p1, p2)$ to zero. For each initial condition, we randomly chose a fixed step size $h$ from the interval $[0, 0.5]$, and integrated the system forward for 100 time steps using a high-order solver. This process was repeated to generate 500 trajectories, each of 100 steps. From the resulting collection of one-step pairs, we randomly shuffled and selected $100, 1000, 5000$ samples to construct training datasets. The neural network consists of 10 layers, each with 100 width. We set the learning rate to $0.0001$ and trained the model for 30000 iterations.

During testing, we followed the same protocol as in training: each test trajectory starts from a randomly chosen initial condition, with angles in $[-0.8, 0.8]$ and momenta set to zero. A fixed step size $h \in [0, 0.5]$ was randomly selected per test and kept constant over 100 steps.

Table 4 shows the average training and test loss with standard deviations for different training dataset sizes. Each result is computed based on 5 independent training runs with different data. For both systems, the training loss consistently converges to a similar small value and the test loss improves as the number of training samples increases.

### D.3   Pendulum Experiment

We conduct experiments on the pendulum (Figure 12). The mathematical pendulum (mass $m = 1$, string length $l = 1$, gravity $g = 1$ ) is a system with the Hamiltonian $H(q, p) = \frac{1}{2}p^2 - \cos(q)$.

Table 4: The values of the train and test loss with different data and initial conditions.

| PROBLEM | TRAIN LOSS | TEST LOSS |
|---|---|---|
| HÉNON-HEILES | | |
| -100 TRAINING DATA | $1.64 \times 10^{-8} \pm 6.09 \times 10^{-9}$ | $3.43 \times 10^{-8} \pm 1.98 \times 10^{-8}$ |
| -1000 TRAINING DATA | $1.83 \times 10^{-8} \pm 2.66 \times 10^{-9}$ | $5.29 \times 10^{-9} \pm 4.45 \times 10^{-9}$ |
| -5000 TRAINING DATA | $1.90 \times 10^{-8} \pm 1.30 \times 10^{-9}$ | $1.58 \times 10^{-9} \pm 1.60 \times 10^{-9}$ |
| 2-PENDULUM | | |
| -100 TRAINING DATA | $2.09 \times 10^{-7} \pm 2.37 \times 10^{-7}$ | $1.22 \times 10^{-5} \pm 1.64 \times 10^{-5}$ |
| -1000 TRAINING DATA | $3.44 \times 10^{-7} \pm 1.11 \times 10^{-7}$ | $6.43 \times 10^{-8} \pm 1.78 \times 10^{-8}$ |
| -5000 TRAINING DATA | $3.09 \times 10^{-7} \pm 3.05 \times 10^{-8}$ | $4.24 \times 10^{-8} \pm 3.52 \times 10^{-8}$ |

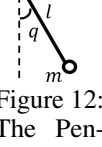

Figure 12: The Pendulum

The position coordinate $q$ denotes the angle from the vertical suspension point and $p = \dot{q}$ denotes the momentum. Although the trigonometric function exists in the Hamiltonian, the integral in the AVF method is analytically computable, and hence the AVF method is available.

In this experiment, we used two sets of data: flow dataset and irregular dataset like the example shown in the Figure 13. The training flow data consists of $N = 15$ points sampled from a trajectory for each fixed time step $h = 0.5$, calculated using a high-order integrator starting from $u^{(0)} = (q^{(0)}, p^{(0)}) = (2, 0)$. The training flow data are pairs of numerical solutions at a certain time step $n$ along with the one at the next time step: $\{(u^{(n-1)}, u^{(n)}) \mid n = 1, ..., N\}$. After training, we use the trained gradient to compute the flow starting at $u^{(N)}$ for 200 steps.

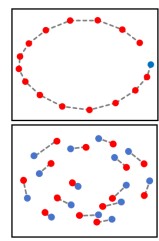

Figure 13: Flow and irregular data with 15 pairs of points

The training irregular data $\{((u^{(i-1)}, h^{(i)}), u^{(i)}) \mid i = 1, ..., I\}$ consists of $I = 100$ pairs of points. $u^{(i-1)}$ are randomly sampled from $[-3, 3] \times [-3, 3]$ and the time steps $h^{(i)}$ are randomly chosen from $[0, 0.8]$. $u^{(i)}$ are numerical solutions at the next step of $u^{(i-1)}$ with the step size $h^{(i)}$. After training, we use the trained gradient to compute the flow starting at $u^{(0)} = (2, 0)$ with $h = 0.5$ for 200 steps.

In Figure 14, we show the additional for the pendulum experiment. The plot in the top-left corner shows the irregular training data. The top-right plot displays the test trajectories of the trained network and numerical integrators during the test. The two middle plots show the values of $q$ and $p$ over time. The bottom-left plot shows the error evolutions over time, while the bottom-right plot illustrates the energy conservation.

In Figure 15, we show the additional results for the pendulum experiment with the irregular data. The plot in the top-left corner shows the irregular training data. The top-right plot displays the test trajectories of the trained network and numerical integrators during the test. The two middle plots show the values of $q$ and $p$ over time. The bottom-left plot shows the error evolutions over time, while the bottom-right plot illustrates the energy conservation.

## D.4 Supplementary figures of the experimental results

**Results for the Hénon-Heiles system** In Figure 16, we show the additional results for the Hénon-Heiles experiment. The plot in the top-left corner shows the positions of the training flow data, while the top-middle displays the values of test $q_1$. The plot in the top-right shows the change in error over time, and the center-right shows energy conservation. The remaining plots display the test trajectories of the ground truth, the solution by the AVF method, that by the AVF method with a half step size, 4 order, and the solution by the proposed method.

In Figure 17, we show the additional results for the Hénon-Heiles experiment with the irregular data. The plot in the top-left corner shows the positions of the randomly selected irregular training data, while the top-middle displays the values of test $q_1$. The plot in the top-right shows the change in error over time, and the center-right shows energy conservation. The remaining plots display the test trajectories of the ground truth, the solution by the AVF method, that by the AVF method with a half step size, 4 order, and the solution by the proposed method.

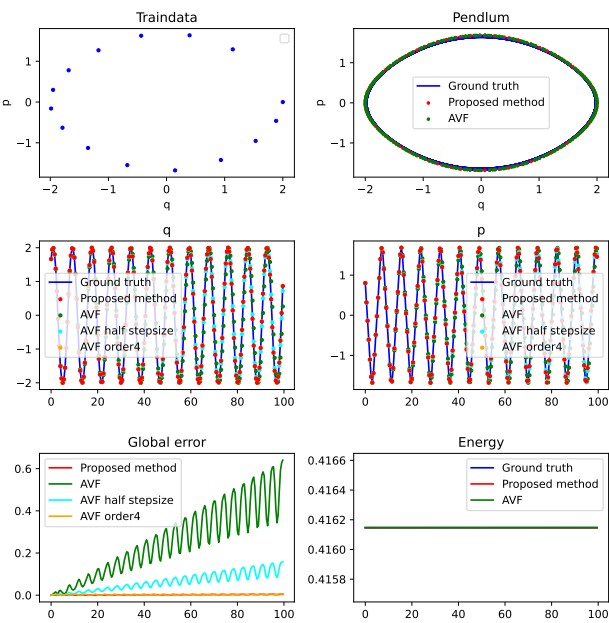

Figure 14: Results for the pendulum experiment

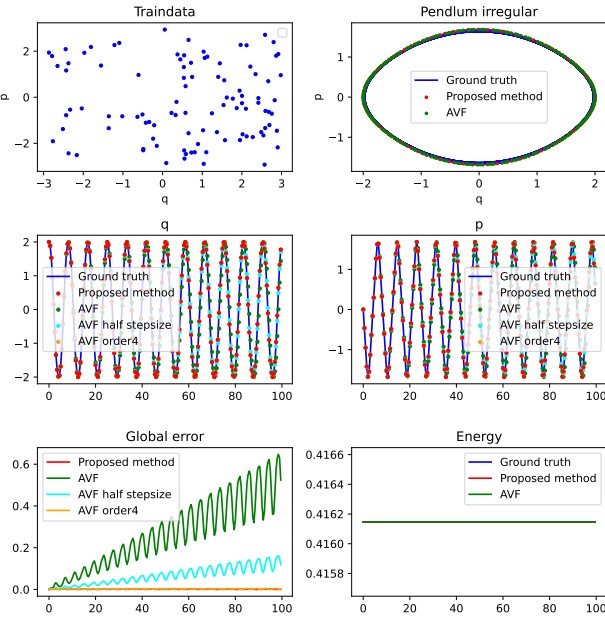

Figure 15: Results for the pendulum experiment with the irregular data

**Results for 2body**    In Figure 18, in addition to the numerical errors and the evolutions of the energy mentioned in the main text, the top-left corner shows the training flow data, while the top-middle display the values of test $q_1$. The plot in the center-right shows energy conservation.

**Results for Double pendulum**    In Figure 19, the plot in the top-left corner shows the positions of the training flow data, while the two plots at the center-right show the evolutions of the errors and the energy over time. The bottom plots display the trajectories of the ground truth, the solution by the AVF method, the solution by the AVF method with a half step size, and the solution by the proposed method.

**Results for the dissipative pendulum**  In Figure 20, in addition to the numerical errors and the evolutions of the energy mentioned in the main text, the top-left corner shows the training flow data, the top-right corner displays the test trajectory after training, the middle display the values of test $q, p$, and the two plots at the bottom correspond to the values of the coordinates over time.

**Results for the Duffing oscillator**  In Figure 21, we show the results for the Duffing oscillator. In addition to the numerical errors and the evolutions of the energy mentioned in the main text, the top-left corner shows the training flow data, the top-right corner displays the test trajectory after training, the middle display the values of test $q, p$, and the two plots at the bottom correspond to the values of the coordinates over time.

**Results for KDV**  We consider this equation on the interval $[0, L]$, $L = 20$ under the periodic boundary condition. We used the training data with the time length of $T = 30$ and the time step size of $\Delta t = 0.01$. The spatial step size was $\Delta x = 0.5$. The initial condition of the training data is constructed by superimposing two solitary wave solutions of the KdV equation: $u_0(x) = u_{sol}(x - x_1, c_1) + u_{sol}(x - x_2, c_2)$, where $u_{sol}(x, c) = \frac{1}{2}c \cdot \text{sech}^2(\frac{1}{2}\sqrt{c} \cdot x)$ with $c_1 = 0.75, x_1 = 0.33L, c_2 = 0.4, x_1 = 0.65L$. The time evolution is computed by solving the KdV equation using the solver with high accuracy, where spatial derivatives were approximated using finite difference methods. For testing, the computation was carried out from the time length $T = 30$, with the start point being at the final state of the training dataset.

In Figure 22, we show the results for the KdV. The plot in the top-left corner shows the positions of the training flow data. The plot in the bottom-left shows the change in error over time, and the bottom-middle shows energy conservation. The remaining plots display the test trajectories of the ground truth, the solution by the AVF method, and the solution by the proposed method.

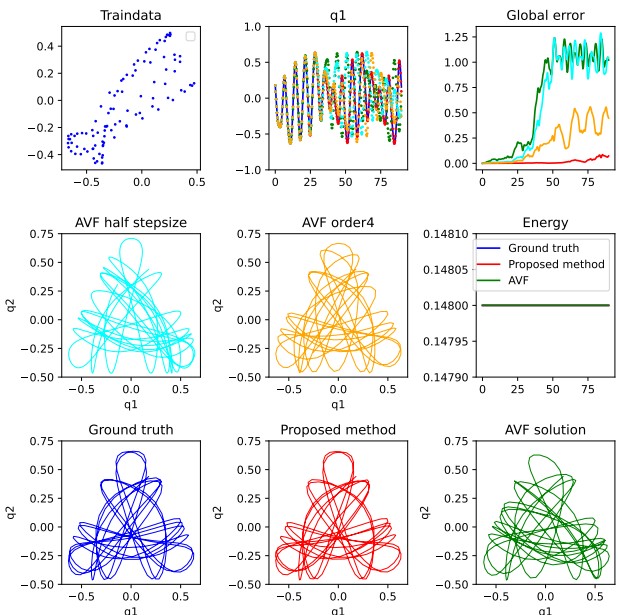

Figure 16: Results for the Hénon-Heiles experiment

.

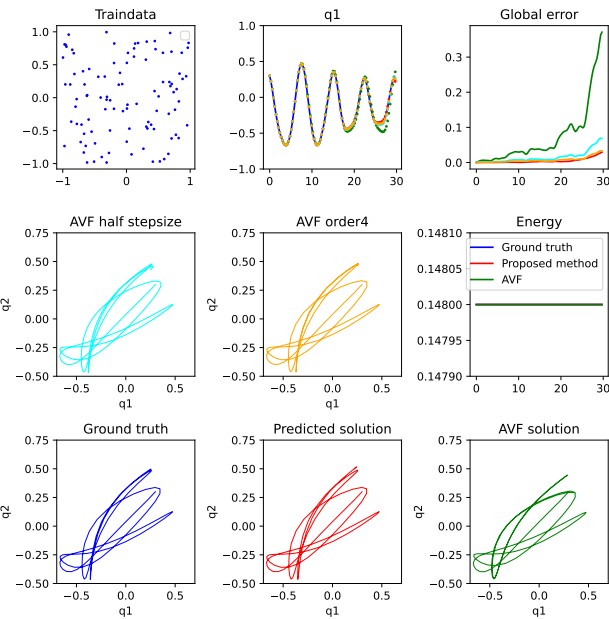

Figure 17: Results for the Hénon-Heiles experiment with the irregular data

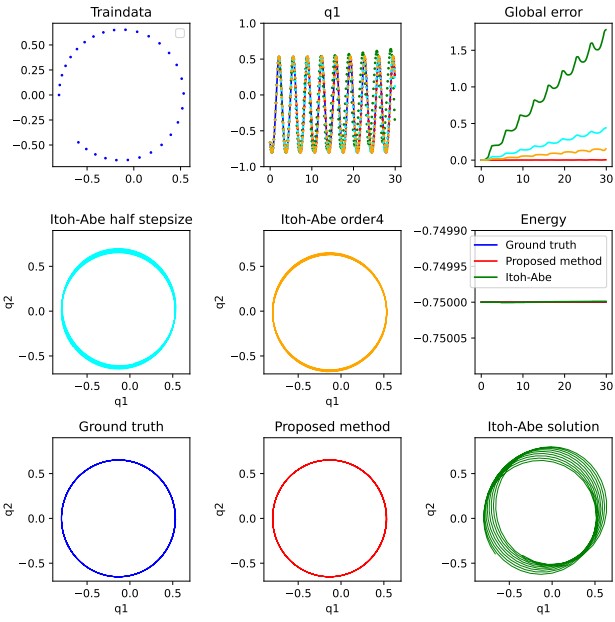

Figure 18: Results for 2body

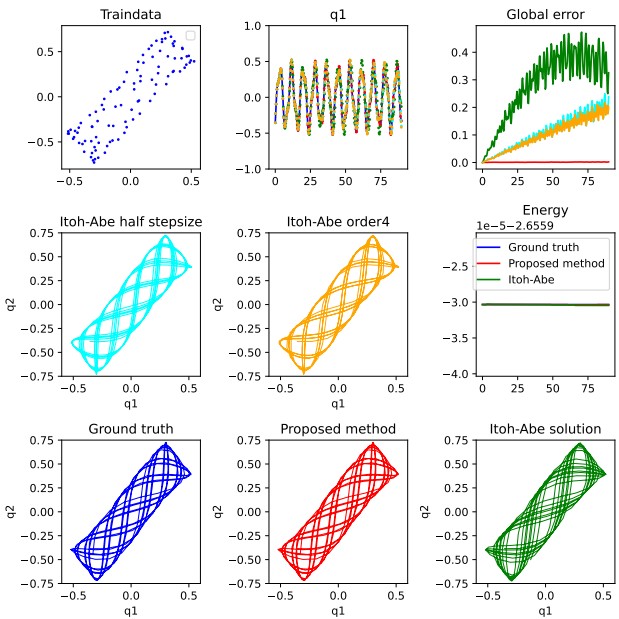

Figure 19: Results for the double pendulum

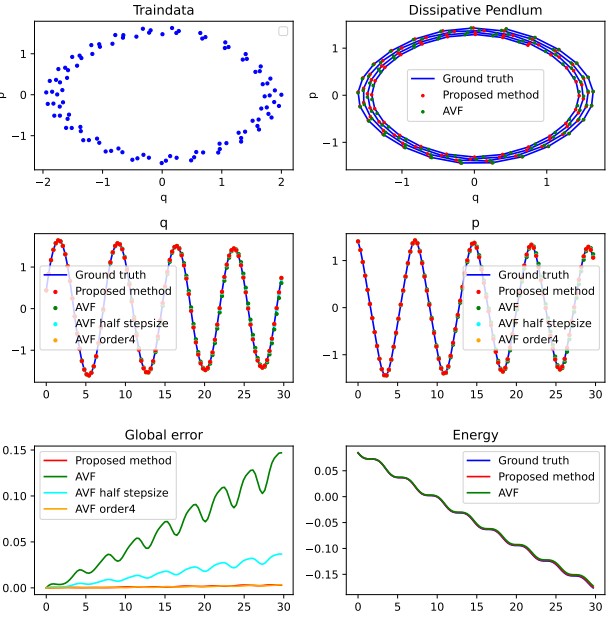

Figure 20: Results for the dissipative pendulum

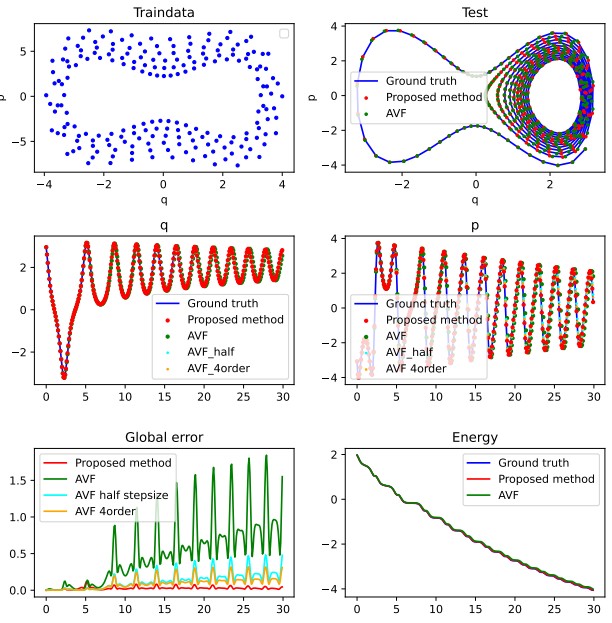

Figure 21: Results for the Duffing oscillator

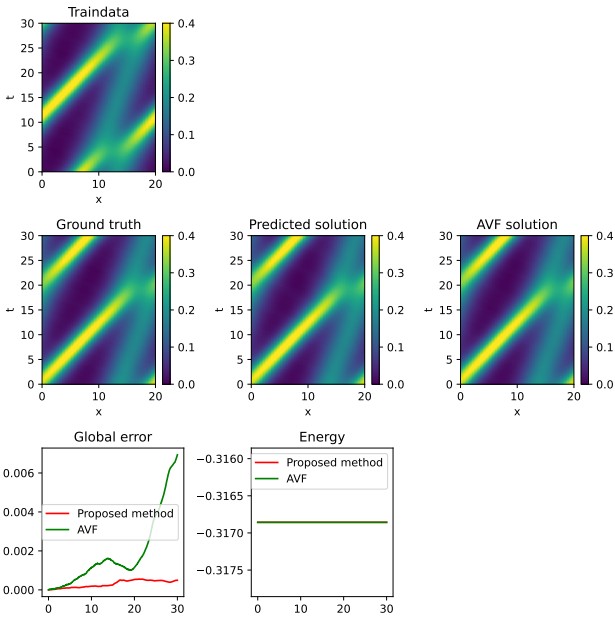

Figure 22: Results for the KdV

### D.5 Comparison with other methods

We implemented the PINNs as in [54] with energy conservation/dissipation regularization terms, using the DeepXDE library, and applied it to three benchmark systems: Hénon–Heiles, double pendulum, and Duffing oscillator. Both our method and the PINNs use the same network architecture (10 layers, width 100), optimizer (Adam, learning rate 0.0001), and 30000 training iterations. We sampled 2000 random collocation points over the spatiotemporal domain.

We also implemented the Time-Reversal Symmetric ODE Network (TRS-ODEN) [28]. For the time-reverse symmetry loss term, we conducted experiments with three different weight values: $\lambda = 10, 1, 0.5$, which are employed in the reference paper. The network architecture (10 layers, width 100), optimizer (Adam, learning rate 0.0001), and 30000 training iterations are kept the same.

To evaluate the performance, we generated a ground-truth reference trajectory for each system using a high-order solver with 100 time steps, starting from the same initial conditions. We then simulated trajectories using both the trained PINNs, TRS-ODEN and our proposed method under the same initial condition. We report the maximum error over the time domain $[0, T]$, defined as the maximum deviation from the ground-truth trajectory. Furthermore, since PINNs tends to degrade over long time horizons, we computed the maximum error accumulated up to each checkpoint ($T = 5, 10, 50, 100$), i.e., the largest deviation from the reference trajectory from step 0 up to the given time step $T$.

**Hénon-Heiles System** We set the initial condition as $u^{(0)} = (q_1^{(0)}, q_2^{(0)}, p_1^{(0)}, p_2^{(0)}) = (0.3, 0.3, -0.2, -0.2)$, and performed simulations over 100 steps with a fixed time step size $h = 0.3$.

**Double Pendulum** We set the initial condition as $u^{(0)} = (q_1^{(0)}, q_2^{(0)}, p_1^{(0)}, p_2^{(0)}) = (\pi/6, \pi/, 0, 0)$, and performed simulations over 100 steps with a fixed time step size $h = 0.3$.

**Duffing Oscillator** We set the initial condition as $u^{(0)} = (q^{(0)}, p^{(0)}) = (2.95, -3.08)$ and performed simulations over 100 steps with a fixed time step size $h = 0.1$.

The maximum errors of both models, measured against the reference solution, are summarized in table 5.

These results in all three systems suggest that our structure-preserving method offers improved accuracy and stability over the other two methods, especially in long-term simulations. This improvement is attributed to its formulation as a numerical integrator designed to respect the system's physical structure.

Table 5: Comparison of Maximum Errors

| TIME STEP | PROPOSED METHOD | PINNS | TRS-ODEN $\lambda = 10$ | $\lambda = 1$ | $\lambda = 0.5$ |
|---|---|---|---|---|---|
| HÉNON-HEILES | | | | | |
| 5 | $1.43 \times 10^{-4}$ | $2.86 \times 10^{-2}$ | $9.11 \times 10^{-3}$ | $3.72 \times 10^{-3}$ | $2.37 \times 10^{-3}$ |
| 10 | $1.54 \times 10^{-4}$ | $9.77 \times 10^{-2}$ | $2.16 \times 10^{-2}$ | $1.24 \times 10^{-2}$ | $4.35 \times 10^{-3}$ |
| 50 | $1.10 \times 10^{-3}$ | $1.48$ | $7.40 \times 10^{-1}$ | $3.16 \times 10^{-1}$ | $5.21 \times 10^{-2}$ |
| 100 | $9.53 \times 10^{-3}$ | $1.50$ | $1.32$ | $3.26 \times 10^{-1}$ | $1.43 \times 10^{-1}$ |
| 2-PENDULUM | | | | | |
| 5 | $1.72 \times 10^{-4}$ | $1.92 \times 10^{-2}$ | $1.32 \times 10^{-2}$ | $1.08 \times 10^{-2}$ | $9.58 \times 10^{-3}$ |
| 10 | $8.28 \times 10^{-4}$ | $5.39 \times 10^{-2}$ | $1.85 \times 10^{-2}$ | $1.18 \times 10^{-2}$ | $9.58 \times 10^{-3}$ |
| 50 | $3.50 \times 10^{-3}$ | $1.98 \times 10^{-1}$ | $1.01 \times 10^{-1}$ | $3.85 \times 10^{-2}$ | $4.37 \times 10^{-2}$ |
| 100 | $6.18 \times 10^{-3}$ | $2.38 \times 10^{-1}$ | $2.64 \times 10^{-1}$ | $7.59 \times 10^{-2}$ | $1.24 \times 10^{-1}$ |
| DUFFING | | | | | |
| 5 | $8.40 \times 10^{-4}$ | $4.20 \times 10^{-1}$ | $1.44 \times 10^{-1}$ | $3.90 \times 10^{-2}$ | $2.96 \times 10^{-2}$ |
| 10 | $9.38 \times 10^{-4}$ | $6.99 \times 10^{-1}$ | $2.06 \times 10^{-1}$ | $3.90 \times 10^{-2}$ | $3.55 \times 10^{-2}$ |
| 50 | $5.18 \times 10^{-2}$ | $4.50$ | $2.86$ | $5.22$ | $3.33$ |
| 100 | $1.12 \times 10^{-1}$ | $4.50$ | $6.40$ | $5.38$ | $5.12$ |

### D.6 Computational Time

We performed additional experiments to re-evaluate the computation time across different systems. In the experiments, we used an Intel Xeon 6900P CPU.

The results, the size of the neural networks used and the number of time steps in the numerical calculations are shown in Table 6. The computation time is generally similar to that of the Hénon-Heiles problem reported in the main body. In particular, the proposed method achieves better accuracy with less computational time than the fourth-order methods. For the KdV equation experiment, the network was made as small as possible without sacrificing accuracy. The network used is quite small, yet the test loss was $1.706 \times 10^{-6}$, which is not much different from the test loss of $1.695 \times 10^{-6}$ when a larger network with 5 layers with width of 200 is used. For other experiments, the computation time can be further reduced by making the network smaller.

Table 6: Comparison of Computational Time (sec)

| Problem | Layer | Width | Steps | Proposed | 2nd Order Methods | 4th Order Methods |
|---------|-------|-------|-------|----------|-------------------|-------------------|
| Pendulum | 5 | 50 | 200 | 0.579 | 0.503 | 1.137 |
| Hénon-Heiles | 10 | 100 | 300 | 2.090 | 1.437 | 3.533 |
| 2-Body | 5 | 50 | 300 | 2.862 | 2.443 | 7.693 |
| 2-Pendulum | 5 | 50 | 300 | 5.768 | 5.197 | 18.043 |
| Duffing | 5 | 50 | 300 | 1.020 | 0.685 | 1.084 |
| KdV | 2 | 30 | 3000 | 6.566 | 1.634 | 8.781 |

