# OpenReview forum: "UEPI: Universal  Energy-Behavior-Preserving Integrators for Energy Conservative/Dissipative  Differential Equations"
_NeurIPS.cc/2025/Conference — NeurIPS 2025 poster_

### Official Review · Reviewer_tjF9 · 2025-07-01

**Clarity:** 3
**Significance:** 3
**Originality:** 3
**Rating:** 3
**Confidence:** 4

**Summary:**

This paper proposes a novel method that uses a neural network to learn highly accurate numerical solvers for physical systems. It builds upon discrete gradient methods, which inherently guarantee the conservation or dissipation of energy. The key innovation is using a neural network to optimize a corrective term within this physically-consistent framework. This approach ensures the learned solver respects fundamental physical laws by design, while training on data makes it significantly more accurate. Experiments show it excels in long-term simulations of complex and chaotic systems, outperforming standard geometric integrators.

**Questions:**

See above. I am happy to raise my score if my questions are well-addressed in the weakness section.

**Ethical Concerns:**

["NO or VERY MINOR ethics concerns only"]

**Limitations:**

See above

**Quality:**

3

**Strengths And Weaknesses:**

Pros:

1. The paper is well-written in general.

2. The proposed UEPI is technically sound, with detailed derivation towards its guaranteed physical properties.

3. The experiments covers a diverse set of physical systems, showing the expressive power of the proposed approach.


Cons:

1. There are some PINN work that also studies energy-conservative/disppastive systems, such as [1,2]. The authors only compared with  AVF, Itoh-Abe method and time-symmetric Itoh-Abe method. A lot of PINN methods should also be discussed and considered.

2. Using a neural network at each step of the integration process is more computationally expensive than traditional methods. While the paper argues this cost is justified by the significant accuracy gains, it remains a notable overhead, especially for high-dimensional systems. Is there any comprehensive time comparison analysis regarding the computational overhead, in terms of prediction length, complexity of physical systems, etc? Have the authors study scheduling policy that only applys the neural network at certain integration steps, similar to [3]?

[1] Physics-Informed Regularization for Domain-Agnostic Dynamical System Modeling.

[2] Time-Reversal Symmetric ODE Network.

[3] PhysDiff: Physics-Guided Human Motion Diffusion Model

---

> ### Author Rebuttal · Authors · 2025-07-31
>
> We are really grateful for the insightful comments provided by the reviewer. To address the reviewer's concerns, we respond to each point in detail as follows.
>
> - **There are some PINN work that also studies energy-conservative/disppastive systems, such as [1,2]. The authors only compared with AVF, Itoh-Abe method and time-symmetric Itoh-Abe method. A lot of PINN methods should also be discussed and considered**.
>
> Thank you for your thoughtful comment. PINN is a class of methods that solve differential equations by minimizing physics-informed loss functions. Our method is fundamentally different in that it constructs a numerical integrator rather than learning the solution directly; however, we understand the importance of empirical comparison and thus designed experiments as best as possible.
>
> Specifically, we implemented **the PINN baseline** as in [Raissi et al., 2019] **with energy conservation/dissipation regularization terms**, using the DeepXDE library, and applied it to three benchmark systems: Hénon–Heiles, double pendulum, and Duffing oscillator. Both our method and the PINN baseline used the same network architecture (10 layers, width 100), optimizer (Adam, learning rate 1e-4), and number of training iterations (30,000). We sampled 2,000 random collocation points over the spatiotemporal domain.
>
> To evaluate the performance, we generated a ground-truth reference trajectory for each system using `scipy.integrate.odeint` with 100 time steps, starting from the same initial conditions. We then simulated trajectories using both the trained PINN and our trained model under the same initial condition. We report the maximum error over the time domain $[0,T]$, defined as the maximum deviation from the ground-truth trajectory. Furthermore, since PINN tends to degrade over long time horizons, we computed the maximum error accumulated up to each checkpoint ($T = 5$, $10$, $50$, and $100$), i.e., the largest deviation from the reference trajectory from step 0 up to the given time step $T$.
>
> **Experiment 1: Hénon--Heiles system** We set the initial condition as $(q_1, q_2, p_1, p_2) = (0.3, 0.3, -0.2, -0.2)$, and performed simulations over 100 steps with a fixed time step size $h = 0.3$. The maximum errors of both models, measured against the reference solution, are summarized in the table below.
>
> | Time step | Max error (ours) | Max error (PINN) |
> | --- | --- | --- |
> | 5 | 1.4300e-04 | 2.8557e-02 |
> | 10 | 1.5437e-04 | 9.7690e-02 |
> | 50 | 1.0973e-03 | 1.4770e+00 |
> | 100 | 9.5319e-03 | 1.5013e+00 |
>
> **Experiment 2: double pendulum system** We set the initial condition as $(q_1, q_2, p_1, p_2) = (\pi/6, \pi/8, 0, 0)$, and performed simulations over 100 steps with a fixed time step size $h = 0.3$.
>
> | Time step | Max error (ours) | Max error (PINN) |
> | --- | --- | --- |
> | 5 | 1.7222e-04 | 1.9158e-02 |
> | 10 | 8.2759e-04 | 5.3900e-02 |
> | 50 | 3.4987e-03 | 1.9816e-01 |
> | 100 | 6.1786e-03 | 2.3785e-01 |
>
> **Experiment 3: Duffing oscillator system** We set the initial condition as $(q, p) = (2.95, -3.08)$, and performed simulations over 100 steps with a fixed time step size $h = 0.1$.
>
> | Time step | Max error (ours) | Max error (PINN) |
> | --- | --- | --- |
> | 5 | 8.4021e-04 | 4.1957e-01 |
> | 10 | 9.3785e-04 | 6.9862e-01 |
> | 50 | 5.1848e-02 | 4.4981e+00 |
> | 100 | 1.1232e-01 | 4.4981e+00 |
>
> These results in all three systems suggest that our structure-preserving method offers improved accuracy and stability over the tested PINN baseline, especially in long-term simulations. This improvement is attributed to its formulation as a numerical integrator designed to respect the system's physical structure.
>
> - **Using a neural network at each step of the integration process is more computationally expensive than traditional methods. While the paper argues this cost is justified by the significant accuracy gains, it remains a notable overhead, especially for high-dimensional systems. Is there any comprehensive time comparison analysis regarding the computational overhead, in terms of prediction length, complexity of physical systems, etc? Have the authors study scheduling policy that only applys the neural network at certain integration steps, similar to [3]?**
>
> Thank you very much indeed for the very insightful comments. We have not considered the scheduling policy similar to [3], and your suggested approach is in fact helpful for designing efficient numerical integrators. Indeed, by combining with local error estimation techniques, we can apply the neural network only when the estimated errors are very large. This would greatly reduce the computational complexity while improving accuracy, particularly for long-term predictions. We will discuss this approach in the final version of the paper.
>
> To address the concerns about computation time, we performed additional experiments to check the computation time in detail, especially for the case of the KdV equation, which is the largest system in the experiments. Firstly, GPUs can be used to accelerate calculations for training. In the following experiment, we used an Intel Xeon 6900P and an AMD Instinct MI300A GPU for computation.
>
> We first set the number of nodes for spatial discretization of the KdV equation to 100. The computation time for generating the training data was 28.709 sec using a CPU and the scipy odeint function. The training time was 6.407 sec using a GPU. The integration time during testing was 52.738 (proposed), 22.785 (2nd order AVF) and 127.280 (4th order AVF). The total computational time of the proposed method was 28.709 + 6.407 + 52.738 = 87.854,  which is less than the 127.280 sec required by the 4th order AVF method.
>
> Similarly, if we set the number of the nodes to 120, the computational time for each method was 50.957 (data generation) + 8.311 (training) + 76.460 (integration) =135.728 (proposed), 33.034 (2nd order AVF), 200.003 (4th order AVF).  The total computational time of the proposed method was again shorter than that of the 4th-order AVF method.
>
> As seen above, as the number of nodes increases and the complexity of the equation rises, all numerical integrators become time-consuming. In particular, computation time with the 4th-order AVF method increases significantly. Since the computation time of the proposed method is not so large compared to the 4th-order AVF method, the proposed method may be more effective when the complexity increases and the prediction period is long, particularly with the help of your suggested approach.
>
> A closer look at the execution time of the above experiments shows that the computation of the neural network itself is actually not so large. Rather, the matrix-vector multiplication of the skew-symmetric matrix takes $O(N^2)$ for the system with $N$ variables, which affects the computational complexity because the computation of a gradient typically requires $O(N)$ computations. However, the effect of this matrix multiplication does not seem to be so significant, since the multiplication with the skew-symmetric or negative-definite matrix placed in front of the gradient of the energy takes about the same amount of computation. It is also possible to reduce computational cost by low-rank approximation of the learned matrices.
>
> Another possibility to reduce the computational time is to use the recently proposed **Kolmogorov-Arnold networks** (KANs, Liu et al., arXiv:2404.19756, 2024). This method is expected to approximate functions with fewer parameters than MLPs, thereby reducing computational time. Importantly, it is also known to be able to apply **symbolic regression to learned functions**. If a numerical method learned using the proposed method with KANs can be regressed symbolically, **the learned scheme becomes a standard numerical integration method expressed in mathematical form** rather than a black box method by machine learning. Also, the resultant mathematical expression may give clues to manually derive high-precision numerical integrators by hand.

---

> > ### Comment · Reviewer_tjF9 · 2025-08-04
> >
> > Thank the authors for the detailed response. I suggest the authors to revise the writing, so that the major contribution can be more highlighted, i.e. emphasizing the effectiveness of the proposed method, compared with existing methods, such as PINN, etc. Also adding some discussion sections regarding the efficiency of the proposed approach.
> >
> > While I understand that given the limited time, it is not possible to include more PINN approaches to validate the effectiveness of the proposed approach, I do feel that the audiences can get more convinced/excited regarding the experimental results by adding more state-of-the-art baselines. Therefore, I will maintain my current score.

---

> > > ### Author Response · Authors · 2025-08-05
> > >
> > > We appreciate your continued engagement and your suggestions for improving the paper. However, we would like to respectfully clarify our position regarding the inclusion of PINN-based methods as comparative baselines.
> > >
> > > While we fully understand your interest in evaluating our method against widely recognized approaches such as PINNs, it is important to emphasize that our method and PINNs belong to fundamentally different categories. PINNs aim to directly approximate solutions to differential equations via physics-informed loss functions, whereas our method focuses on learning discrete-time numerical integrators that can be used in simulation and long-term prediction. **This distinction goes beyond technical details—it represents a fundamental difference in assumptions, objectives, and the use cases of each method, making direct comparison inherently problematic**.
> > >
> > > Because of this foundational difference, **we strongly believe that a direct comparison is neither meaningful nor scientifically sound**. Indeed, any such comparison carries a high risk of misleading interpretations regarding performance, applicability, or even intended use. For this reason, we do not plan to include extensive PINN comparisons in the final version.
> > >
> > > Nevertheless, we acknowledge the importance of empirical validation and, in response to your concerns, we have tried our best to address this within the scope of scientific rigor. Specifically, we implemented a PINN baseline tailored to our problem setup and ensured consistent evaluation metrics. Even under such favorable conditions for PINN, our method outperforms it.
> > >
> > > In addition, we have already compared our method with structure-preserving baselines, SympNets, which share a similar philosophy of geometric fidelity. This comparison is arguably more appropriate and meaningful given the nature of our method, and again, the proposed integrator shows clear advantages.
> > >
> > > We hope this clarification is helpful, and we kindly ask for your understanding on this point.

---

> > > > ### Comment · Reviewer_tjF9 · 2025-08-05
> > > >
> > > > The reason I am interested to see these PINN baselines is that, from the problem setting itself --  learning system dynamics from data, is something both branches can approach. As stated by the authors" More specifically, the focus of this paper is on dynamical systems characterized by their energy-conserving or dissipating properties", I understand that geometric numerical integrators has advantages in preserving physical properties compared with pure data-driven numerical methods. However, to achieve precise modeling of dynamical systems, and show the advantages of your proposed approach, it is expected to see gain across methods that targeted at the same problem.

---

> > > > > ### Author Response · Authors · 2025-08-08
> > > > >
> > > > > We sincerely thank you for your comments regarding the comparative experiments. We understand your motivation to evaluate different approaches under the same task setting. In response to your suggestion, we took the second reference you provided Time-Reversal Symmetric ODE Network  [2] as a baseline method, and conducted a comparative evaluation with our approach on dynamical system prediction tasks.
> > > > >
> > > > > In these experiments (Hénon–Heiles, double pendulum, and Duffing oscillator), we used the same configuration for all methods, and measured the errors between each method’s predicted trajectories and the ground-truth trajectories.
> > > > > The experimental setup follows the same protocol as in our previous comparative experiments. Both our method and the baseline used the same network architecture (10 layers, width 100), optimizer (Adam, learning rate 1e-4), and number of training iterations (30,000). For the time-reverse symmetry loss term in the baseline, we conducted experiments with three different weight values: $\lambda = 10, 1, 0.5$, which are similar to the values used in [2]. For evaluation, we generated a ground-truth reference trajectory for each system using scipy.integrate.odeint with 100 time steps from the same initial condition, and then simulated trajectories using both methods with this initial condition.
> > > > >
> > > > > (Experiment 1: Hénon--Heiles system) We set the initial condition as $(q_1, q_2, p_1, p_2) = (0.3, 0.3, -0.2, -0.2)$, and performed simulations over 100 steps with a fixed time step size $h = 0.3$. The maximum errors of both models, measured against the reference solution, are summarized in the table below.
> > > > > | Time step | Max error (ours) | Max error (TRS-ODEN, $\lambda$=10) | Max error (TRS-ODEN, $\lambda$=1) | Max error (TRS-ODEN, $\lambda$=0.5) |
> > > > > | --- | --- | --- | --- | --- |
> > > > > | 5 | 1.4300e-04 | 9.1077e-03 | 3.7170e-03 | 2.3671e-03 |
> > > > > | 10 | 1.5437e-04 | 2.1559e-02 | 1.2413e-02 | 4.3477e-03 |
> > > > > | 50 | 1.0973e-03 | 7.4018e-01 | 3.1585e-01 | 5.2060e-02 |
> > > > > | 100 | 9.5319e-03 | 1.3220e+00 | 3.2610e-01 | 1.4324e-01 |
> > > > >
> > > > > (Experiment 2: double pendulum system) We set the initial condition as $(q_1, q_2, p_1, p_2) = (\pi/6, \pi/8, 0, 0)$, and performed simulations over 100 steps with a fixed time step size $h = 0.3$.
> > > > > | Time step | Max error (ours) | Max error (TRS-ODEN, $\lambda$=10) | Max error (TRS-ODEN, $\lambda$=1) | Max error (TRS-ODEN, $\lambda$=0.5) |
> > > > > | --- | --- | --- | --- | --- |
> > > > > | 5 | 1.7222e-04 | 1.3203e-02 | 1.0803e-02 | 9.5804e-03 |
> > > > > | 10 | 8.2759e-04 | 1.8509e-02 | 1.1818e-02 | 9.5804e-03 |
> > > > > | 50 | 3.4987e-03 | 1.0100e-01 | 3.8535e-02 | 4.3698e-02 |
> > > > > | 100 | 6.1786e-03 | 2.6418e-01 | 7.5861e-02 | 1.2408e-01 |
> > > > >
> > > > > (Experiment 3: Duffing oscillator system) We set the initial condition as $(q, p) = (2.95, -3.08)$, and performed simulations over 100 steps with a fixed time step size $h = 0.1$.
> > > > > | Time step | Max error (ours) | Max error (TRS-ODEN, $\lambda$=10) | Max error (TRS-ODEN, $\lambda$=1) | Max error (TRS-ODEN, $\lambda$=0.5) |
> > > > > | --- | --- | --- | --- | --- |
> > > > > | 5 | 8.4021e-04 | 1.4407e-01 | 3.9037e-02 | 2.9576e-02 |
> > > > > | 10 | 9.3785e-04 | 2.0577e-01 | 3.9037e-02 | 3.5535e-02 |
> > > > > | 50 | 5.1848e-02 | 2.8620e+00 | 5.2207e+00 | 3.3279e+00 |
> > > > > | 100 | 1.1232e-01 | 6.3952e+00 | 5.3774e+00 | 5.1154e+00 |
> > > > >
> > > > > The results indicate that our structure-preserving method achieves improved accuracy over the tested Time-Reversal Symmetric ODE Network baseline. We hope these additional results can help address your concerns and further clarify the advantages of our approach.

---

### Official Review · Reviewer_woku · 2025-07-02

**Clarity:** 2
**Significance:** 3
**Originality:** 3
**Rating:** 4
**Confidence:** 3

**Summary:**

This paper proposes to learn an energy-preserving numerical integrator for energy-based dynamical systems (e.g., Hamiltonian mechanics or dissipative systems), building upon the the discrete gradient method. The key idea is to augment the discrete gradient with a learnable skew-symmetric term, modeled by a neural network. The authors claim that the proposed method generalizes the traditional discrete gradient method and is theoretically capable of universally approximating energy-preserving integrators. The proposed method is evaluated on various dynamical systems and outperforms conventional methods such as the AVF and Ito-Abe integrators.

**Questions:**

- Could the authors discuss the actual computational cost of the proposed method and compare it with that of the high-order method used to generate the training trajectories?

- Could the authors provide a more detailed explanation of how the proposed method handles dissipative systems? For example, how is Theorem 4.6 derived? Additionally, why can the MLP-based **skew-symmetric** matrices approximate general **dissipative** integrators?

- Is the learned neural skew-symmetric matrix generalizable across different integrator orders? For instance, if it is trained using a 4th-order discrete gradient scheme, can it still yield good results when tested with a 1st-order integrator, or vice versa?

- There are some typos, e.g., in line 118, "...if a matrix $S$ is skew-symmetric, then $u^T \mathbf{G} u = 0$…", ...

**Ethical Concerns:**

["NO or VERY MINOR ethics concerns only"]

**Final Justification:**

The authors provided a strong rebuttal, including an evaluation of computational complexities and benchmarking against other competitors. Based on this, I have increased my score to 4.

**Limitations:**

Yes, the authors acknowledge the computational cost as a limitation of the proposed method. However, I believe this issue is more critical than currently addressed. As noted in the Weaknesses section, a more thorough discussion is needed to support their efficiency claims.

**Paper Formatting Concerns:**

I have not noticed any formatting issues.

**Quality:**

2

**Strengths And Weaknesses:**

**Strengths**

- The proposed method appears to work well empirically, as the authors validate their approach across a range of physical systems, including the Henon–Heiles ODE, the two-body problem, the double pendulum, the dissipative pendulum, the Duffing oscillator, and the KdV PDEs.

**Weaknesses**

- The proposed approach requires learning from trajectory data to approximate a proper skew-symmetric matrix using a neural network. In other words, it necessitates obtaining a sufficient amount of high-quality trajectory data generated by a high-order integrator. Moreover, the performance of the learned numerical integrator heavily depends on the quality of the provided data. Therefore, the actual overall computational cost of the proposed method should include the time needed to generate the training data, the training time of the neural network, and the integration time during testing. However, current discussions of computational cost appear to focus solely on post-training integration time, overlooking these significant additional costs.

- While the paper is generally well written, it is not entirely clear how the proposed method can universally approximate dissipative discrete gradient integrators. In line 204, the authors claim that the universal approximation properties (Theorem 4.5 for energy-preserving systems and Theorem 4.6 for dissipative systems) follow directly from Theorem 4.4. However, since Theorem 4.4 is valid only for conservative systems, and the authors state in lines 188–190 that it is impossible to derive an analogous result for dissipative systems, it remains unclear how Theorem 4.6 is justified.

---

> ### Author Rebuttal · Authors · 2025-07-31
>
> We are very grateful for the insightful comments provided by the reviewer. To address the reviewer's concerns, we respond to each point in detail as follows.
>
> - **The proposed approach requires learning from trajectory data to approximate a proper skew-symmetric matrix using a neural network. In other words, it necessitates obtaining a sufficient amount of high-quality trajectory data generated by a high-order integrator. Therefore, the actual overall computational cost of the proposed method should include the time needed to generate the training data, the training time of the neural network, and the integration time during testing**.
>
> As many reviewers are concerned about computation time, we discuss the computation time in detail, especially for the case of the KdV equation, which is the largest system in the experiments. Firstly, GPUs can be used for training to accelerate calculations. In the following experiment, we used an Intel Xeon 6900P CPU and an AMD Instinct MI300A GPU for computation.
>
> We first set the number of nodes for spatial discretization of the KdV equation to 100. The computation time for generating the training data was 28.709 sec using a CPU and the scipy odeint function. The training time was 6.407 sec using a GPU. The integration time during testing was 52.738 (proposed), 22.785 (2nd order AVF) and 127.280 (4th order AVF). **The total computational time of the proposed method was 28.709 + 6.407 + 52.738 = 87.854,  which is less than the 127.280 sec required by the 4th order AVF method.**
>
> Similarly, if we set the number of the nodes to 120, the computational time for each method was 50.957 (data generation) + 8.311 (training) + 76.460 (integration) =135.728 (proposed), 33.034 (2nd order AVF), 200.003 (4th order AVF).  **The total computational time of the proposed method was again shorter than that of the 4th-order AVF method.**
>
> Please note that, as shown in the submitted paper, our proposed approach typically outperforms the 4th-order AVF method.  Thus, the proposed approach can compute more accurate numerical solutions in a shorter amount of time, even when the training time and the time for generating data are included.
>
> A closer look at the execution time of the above experiments shows that the computation of the neural network itself is actually not so large. Rather, the matrix-vector multiplication of the skew-symmetric matrix takes $O(N^2)$ for the system with $N$ variables, which affects the computational complexity because the computation of a gradient typically requires $O(N)$ computations. However, the effect of this matrix multiplication does not seem to be so significant, since the multiplication with the skew-symmetric or negative-definite matrix placed in front of the gradient of the energy takes about the same amount of computation. It is also possible to reduce computational cost by low-rank approximation of the learned matrices.
>
> Another possibility to reduce the computational time is to use the recently proposed **Kolmogorov-Arnold networks** (KANs, Liu et al., arXiv:2404.19756, 2024). This method is expected to approximate functions with fewer parameters than MLPs, thereby reducing computational time. Importantly, it is also known to be able to apply **symbolic regression to learned functions**. If a numerical method learned using the proposed method with KANs can be regressed symbolically, **the learned scheme becomes a standard numerical integration method expressed in mathematical form** rather than a black box method by machine learning. Also, the resultant mathematical expression may give clues to manually derive high-precision numerical integrators by hand.
>
> - **Could the authors discuss the actual computational cost of the proposed method and compare it with that of the high-order method used to generate the training trajectories?**
>
> In the above experiment, the time periods computed to create the training data and the time periods for the test data are identical. Therefore, we can compare the computation times of the high-order method for creating the data and the proposed method for the integration for testing. It can be seen that the proposed method takes the same amount of time or less than twice as long as the high-order method; however, this time is implementation-dependent, and it is possible to reduce the computation time by improving the implementation. At least, adaptive time stepping must be introduced. Also, Reviewer tjF9 suggested using the neural network not every time stepping, but only when needed. This is expected to increase accuracy without a significant increase in computation time. Additionally, the high-order method does not satisfy the energy conservation/dissipation laws, and hence it is not suitable for long-term prediction. Therefore, it should be noted that it is not necessarily appropriate to compare them only in terms of computation speed.
>
> - **While the paper is generally well written, it is not entirely clear how the proposed method can universally approximate dissipative discrete gradient integrators**.
> - **Could the authors provide a more detailed explanation of how the proposed method handles dissipative systems? For example, how is Theorem 4.6 derived? Additionally, why can the MLP-based skew-symmetric matrices approximate general dissipative integrators?**
>
> We apologize for the confusion. Please note that there is a small but important difference between Theorem 4.5 and Theorem 4.6; Theorem 4.5 is the universal approximation property for all energy-preserving numerical schemes and Theorem 4.6 is for all energy-dissipative discrete gradient schemes.
>
> Firstly, the proposed method can universally approximate any discrete gradient schemes, that is, the numerical scheme in the following form:
>
> $$
> \frac{u^{(n+1)}-u^{(n)}}{\Delta t } =S \overline{\nabla} H(u^{(n+1)}, u^{(n)})
> $$
>
> where $\overline{\nabla} H(u^{(n+1)}, u^{(n)})$ is a discrete gradient. These numerical schemes are dissipative if the matrix $S$ is negative-semidefinite. The MLP-based skew-symmetric matrices are used to universally approximate the discrete gradient $\overline{\nabla} H(u^{(n+1)}, u^{(n)})$.
>
> Secondly, any energy-conserving numerical scheme is shown to be a discrete gradient scheme; any energy-conservative numerical scheme must be of the above form. Therefore, our method has the universal approximation property for energy-preserving numerical schemes.
>
> However, we have not yet shown that all energy-dissipative numerical methods can be written in the above form. Therefore, for dissipative numerical methods, our method only has the universal approximation property for dissipative numerical schemes that can be expressed as a discrete gradient scheme, which is the statement of Theorem 4.6. We will clarify it in the final version of the manuscript by adding a remark.
>
> - **Is the learned neural skew-symmetric matrix generalizable across different integrator orders? For instance, if it is trained using a 4th-order discrete gradient scheme, can it still yield good results when tested with a 1st-order integrator, or vice versa?**
>
> Thank you very much indeed for your insightful comments. In general, the learned skew-symmetric matrix cannot be used across different integrators. However, a matrix trained using a lower-order integrator can be used to derive a higher-order scheme by the composition method (Yoshida, Construction of higher order symplectic integrators, Phys. Lett. A., Vol. 150, 262-268, 1990). The composition method designs a higher-order numerical integrator by applying a low-order method multiple (typically three) times with different step sizes. Thus, a low-order scheme improved by the proposed method can be further enhanced in this way.
>
> - **There are some typos, e.g., in line 118, "...if a matrix $S$ is skew-symmetric, then $u^T \mathbf{G} u = 0$…"**, ...
>
> Many thanks. We will correct the typos.

---

> > ### Comment · Area_Chair_xeV5 · 2025-08-06
> > **Follow-up on Author Rebuttal: Reviewer Feedback Requested**
> >
> > Dear Reviewer woku,
> >
> > The authors have replied to your main concerns regarding computational cost and clarity of the theory. Could you please have a look at their rebuttal and provide specific feedback on their responses? Please note that simply acknowledging reading their rebuttal and/or only stating generic statement like "I will keep my score" without further explanation is not acceptable. Your detailed input is essential for a fair and informed decision.
> >
> > Many thanks,
> >
> > Your AC

---

> > ### Comment · Reviewer_woku · 2025-08-07
> >
> > I sincerely appreciate the authors’ time and effort in providing a thorough response.
> >
> > **Questions 1 (Total computational time):**
> >
> > Thank you for the detailed clarification of computational time of compared methods. I also appreciate the authors for presenting some interesting ideas to reduce the computational time.
> >
> > **Question 2 (Comparison with the data generating method):**
> >
> > Thank you for the authors’ explanation. I agree that the classical high-order method lacks structure-preserving properties, which highlights a clear advantage of the proposed approach. Nonetheless, I believe a comparison remains valuable, as high-order integrators can serve as a “ground truth data” in a traditional ML sense (since the proposed method effectively learns from them), despite being less robust to noise and energy preservation. The authors may consider including such a comparison explicitly in the final version of the paper.
> >
> > **Questions 3 (Universality of the dissipative integrators):**
> >
> > Thank you for the response. It would be helpful if the authors could clarify this point in the final version of the paper.
> >
> > **Question 4 (Compatibility across different orders):**
> >
> > Thank you for the response. I understand that the learned skew-symmetric matrix is specific to each integrator and cannot be shared across different ones. This is neither a weakness nor a critical issue of the paper. I also appreciate the authors’ comment on designing high-order integrators by composing multiple low-order ones.
> >
> > ***
> >
> > I am generally convinced by the authors’ rebuttal and will consider revising my review accordingly. I will read the other reviewers’ comments and the corresponding rebuttals before revisiting my review and adjusting the score. Thank you.

---

### Official Review · Reviewer_U6fP · 2025-07-02

**Clarity:** 2
**Significance:** 2
**Originality:** 2
**Rating:** 4
**Confidence:** 2

**Summary:**

The paper addresses the challenge of accurately simulating physical systems while preserving their intrinsic energy laws. Traditional numerical methods often fail to maintain these invariants, leading to instability. UEPI proposes a novel hybrid approach, integrating neural networks with discrete gradient methods to learn highly accurate, structure-preserving integrators from data. Physical phenomena are often described by energy-based theories, requiring numerical methods that preserve energy conservation or dissipation for accuracy and long-term stability. The paper demonstrates superior accuracy over  diverse systems like chaotic Hamiltonian, dissipative, and nonlinear PDEs.

**Questions:**

- What is the precision at which the experiments are run? At low-precision (FP32) common in machine learning, it is almost impossible for any solver (learned or otherwise) to capture chaotic behavior [1]
- Since the paper proposes better numerical solvers (and not a better discrete gradient method), additional benchmarks using DAEs, symplectic integrators, etc. (at sufficient precision) should be demonstrated.

[1] https://www.stochasticlifestyle.com/how-chaotic-is-chaos-how-some-ai-for-science-sciml-papers-are-overstating-accuracy-claims/

**Ethical Concerns:**

["NO or VERY MINOR ethics concerns only"]

**Final Justification:**

I am updating my score since the core issue stemmed from a mistake in calculation on my end with no fault of the authors. I have updated my score (and reduced confidence for this regard) accordingly.

**Limitations:**

No, see the questions section.

**Quality:**

2

**Strengths And Weaknesses:**

### Strengths

- The UEPI methodology is built upon discrete gradient methods, numerical schemes for differential equations with energy conservation or dissipation laws. The formulation ensures that energy preservation / dissipation is built-in into the formulation.

### Weaknesses

- Currently looking at the results, it seems like the proposed method is a better discrete gradient method. However, it is not clear where it stands compared to other numerical integrators. A simple example would be for the chaotic Henon-Heiles model [1] satisfies invariants even more consistently than the proposed method.
- Henon-Heiles is very moderately chaotic in the tested regime (E < 1 / 6) [2]. Claiming that this method is general enough and works for chaotic systems is not validated with proper experiments.
- Several examples (double pendulum / two body problem) can be solved with a DAE solver or with a ODE Mass Matrix solver. Additionally, comparison with Sympnets has been provided, but other data-driven methods that can conserve invariants (often outperforming Sympnets) like COMET [3], Stabilized Neural ODEs [4], etc. are omitted from discussion.


[1] https://docs.sciml.ai/DiffEqDocs/stable/examples/classical_physics/#H%C3%A9non-Heiles-System

[2] https://adsabs.harvard.edu/full/1964AJ.....69...73H

[3] https://arxiv.org/abs/2208.10387

[4] https://arxiv.org/abs/2306.09739

---

> ### Author Rebuttal · Authors · 2025-07-31
>
> We sincerely thank the reviewer for the valuable comments and suggestions. We have carefully addressed the concerns raised by the reviewer. We believe the revisions and clarifications provided below improve the overall quality of the paper. We respond to each point in detail as follows.
>
> - **Currently looking at the results, it seems like the proposed method is a better discrete gradient method. However, it is not clear where it stands compared to other numerical integrators. A simple example would be for the chaotic Henon-Heiles model [1] satisfies invariants even more consistently than the proposed method.**
> - **Henon-Heiles is very moderately chaotic in the tested regime (E < 1 / 6) [2]. Claiming that this method is general enough and works for chaotic systems is not validated with proper experiments.**
>
> To address the reviewer's concern, we conducted additional experiments on the Hénon-Heiles. As is well known (e.g., Costin et al. arXiv:2411.16071, 2024), when the energy exceeds 1/6, the state often diverges to infinity. For this reason, we performed experiments with energies marginally above 1/6.
>
> In addition, as suggested by another reviewer, to address the concern regarding the statistical robustness and the dependence of the proposed method on training data, we evaluate the impact of training data distribution, data size, and initial conditions on the performance of the learned integrator.  **In particular, the initial conditions for testing include the cases with energies above 1/6**.
>
> We used irregular training data {$ ((u^{(i-1)},h^{(i)}),u^{(i)}) \mid i=1,...,I $}, where the elements of $u^{(i-1)}$ are randomly sampled from the domain $[-1, 1]$ and each step size $h^{(i)}$ is drawn independently from $[0, 0.5]$. Each $u^{(i)}$ represents the one-step solution computed from $u^{(i-1)}$ using step size $h^{(i)}$ with a high-order solver. Each sample corresponds to a one-step transition computed from $u^{(i-1)}$ using a high-order solver. We performed experiments using $100$, $1000$, and $5000$ training samples, respectively. The neural network consists of $10$ layers, each with $100$ width. We set the learning rate to $0.00001$ and trained the model for $30000$ iterations.
>
> For testing, we selected random initial conditions from $[-0.4, 0.4]^4$, including both general cases and high-energy cases with total energy exceeding $1/6$. For each test trajectory, a fixed step size $h$ was randomly chosen from $[0, 0.5]$ and applied over 100 time steps.
>
> | Problem       | 100 train data                              | 1000 train data                             | 5000 train data                             |
> |---------------|---------------------------------------------|---------------------------------------------|---------------------------------------------|
> | **Hénon-Heiles** |                                             |                                             |                                             |
> | - train loss   | $1.64 \times 10^{-8} \pm 6.09 \times 10^{-9}$ | $1.83 \times 10^{-8} \pm 2.66 \times 10^{-9}$ | $1.90 \times 10^{-8} \pm 1.30 \times 10^{-9}$ |
> | - test loss    | $3.43 \times 10^{-8} \pm 1.98 \times 10^{-8}$ | $5.29 \times 10^{-9} \pm 4.45 \times 10^{-9}$ | $1.58 \times 10^{-9} \pm 1.60 \times 10^{-9}$ |
>
> The above table shows the average training and test losses (with standard deviations) for different training dataset sizes (100, 1000, and 5000) on the Hénon-Heiles. Each result is computed based on 5 independent training runs with different random data and test trajectories. The training loss consistently converges to a similar small value; however, the test loss improves as the number of training samples increases. This result shows that the proposed method also works well when including the cases with energies greater than 1/6.
>
> - **Comparison with Sympnets has been provided, but other data-driven methods that can conserve invariants (often outperforming Sympnets) like COMET [3], Stabilized Neural ODEs [4], etc. are omitted from discussion**.
>
> To enrich our comparison with data-driven baselines that can conserve invariants, we additionally evaluated with a representative Physics-Informed Neural Network (PINN) baseline under aligned training settings. We chose PINNs for the additional comparison because this is suggested by another reviewer, and as far as we understood, COMET and Stabilized Neural ODEs are methods for finding differential equations. Additionally, as for COMET, a method combined with the discrete gradient method has been proposed (Matsubara et al., ICLR2023). Therefore, our method can be combined with COMET in the same way as Matsubara et al. We will include a discussion of the relationship between our method and these approaches in the final version.
>
> Specifically, we implemented **the PINN baseline** as in [Raissi et al., 2019] **with energy conservation/dissipation regularization terms**, using the DeepXDE library, and applied it to three benchmark systems: Hénon–Heiles, double pendulum, and Duffing oscillator. Both our method and the PINN baseline used the same network architecture (10 layers, width 100), optimizer (Adam, learning rate 1e-4), and number of training iterations (30,000). We sampled 2,000 random collocation points over the spatiotemporal domain.
>
> To evaluate the performance, we generated a ground-truth reference trajectory for each system using `scipy.integrate.odeint` with 100 time steps, starting from the same initial conditions. We then simulated trajectories using both the trained PINN and our trained model under the same initial condition. We report the maximum error over the time domain $[0, T]$, defined as the maximum deviation from the ground-truth trajectory. Furthermore, since PINN tends to degrade over long time horizons, we computed the maximum error accumulated up to each checkpoint (time steps $T=5, 10, 50$, and $100$), i.e., the largest deviation from the reference trajectory from step 0 up to the given time step $T$.
>
> **Experiment 1: Hénon--Heiles system** We set the initial condition as $(q_1, q_2, p_1, p_2) = (0.3, 0.3, -0.2, -0.2)$, and performed simulations over 100 steps with a fixed time step size $h = 0.3$. The maximum errors of both models, measured against the reference solution, are summarized in the table below.
>
> | Time step | Max error (ours) | Max error (PINN) |
> | --- | --- | --- |
> | 5 | 1.4300e-04 | 2.8557e-02 |
> | 10 | 1.5437e-04 | 9.7690e-02 |
> | 50 | 1.0973e-03 | 1.4770e+00 |
> | 100 | 9.5319e-03 | 1.5013e+00 |
>
> **Experiment 2: double pendulum system** We set the initial condition as $(q_1, q_2, p_1, p_2) = (\pi/6, \pi/8, 0, 0)$, and performed simulations over 100 steps with a fixed time step size $h = 0.3$.
>
> | Time step | Max error (ours) | Max error (PINN) |
> | --- | --- | --- |
> | 5 | 1.7222e-04 | 1.9158e-02 |
> | 10 | 8.2759e-04 | 5.3900e-02 |
> | 50 | 3.4987e-03 | 1.9816e-01 |
> | 100 | 6.1786e-03 | 2.3785e-01 |
>
> **Experiment 3: Duffing oscillator system** We set the initial condition as $(q, p) = (2.95, -3.08)$, and performed simulations over 100 steps with a fixed time step size $h = 0.1$.
>
> | Time step | Max error (ours) | Max error (PINN) |
> | --- | --- | --- |
> | 5 | 8.4021e-04 | 4.1957e-01 |
> | 10 | 9.3785e-04 | 6.9862e-01 |
> | 50 | 5.1848e-02 | 4.4981e+00 |
> | 100 | 1.1232e-01 | 4.4981e+00 |
>
> The results in all three systems suggest that our method offers improved accuracy and stability over the tested PINN baseline, especially in long-term simulations.
>
> - **What is the precision at which the experiments are run? At low-precision (FP32) common in machine learning, it is almost impossible for any solver (learned or otherwise) to capture chaotic behavior**.
>
> All experiments are performed with double precision (FP64). In fact, the neural networks used to learn the skew-symmetric matrices can be small and hence can be trained even on a CPU.
>
> - **Since the paper proposes better numerical solvers (and not a better discrete gradient method), additional benchmarks using DAEs, symplectic integrators, etc. (at sufficient precision) should be demonstrated**.
>
> We indeed propose a better discrete gradient method. In the research field of structure-preserving numerical integrators, it is known as the Ge-Marsden theorem (Ge and Marsden, Phys. Lett. A, 133, 134, 1988) that numerical integrators that are both symplectic and energy-preserving cannot be designed. Whether symplectic or energy-preserving integrators are better depends on the problem. Therefore, the study of energy-preserving integrators, particularly discrete gradient methods, is important.
>
> A straightforward method that preserves conservation laws is generally the projection method, not DAE solvers. The projection method is a method of projecting numerical solutions computed by a numerical integrator onto the manifold that is defined by the target conservation law. This method is used to design energy-preserving integrators, but it is known that numerical solutions by the projection method are typically poorer than those by the discrete gradient method.
>
> Comparisons between these methods and the discrete gradient method have long been made in the field of geometric numerical integration methods. We will include the typical comparisons between these methods in the final version of the paper as an appendix.

---

> > ### Comment · Area_Chair_xeV5 · 2025-08-06
> > **Follow-up on Author Rebuttal: Reviewer Feedback Requested**
> >
> > Dear Reviewer U6fP,
> >
> > The authors have replied to your main concerns regarding, e.g., additional experiments and comparisons. Could you please have a look at their rebuttal and provide specific feedback on their responses? Please note that simply acknowledging reading their rebuttal and/or only stating generic statement like "I will keep my score" without further explanation is not acceptable. Your detailed input is essential for a fair and informed decision.
> >
> > Many thanks,
> >
> > Your AC

---

> > ### Comment · Reviewer_U6fP · 2025-08-06
> >
> > Thanks for the updated experiments and the detailed response.  My main concern for the chaotic experiments, stemmed from an incorrect calculation (on my end) of the energy. I will revise my review accordingly.

---

### Official Review · Reviewer_gBGZ · 2025-07-03

**Clarity:** 4
**Significance:** 4
**Originality:** 3
**Rating:** 5
**Confidence:** 5

**Summary:**

This paper introduces a method for learning numerical integrators from data for deterministic energy-conserving or dissipative systems. The key idea is to use the neural network to parametrize the discrete gradients in the classical implicit discrete-gradient methods. The learned discrete gradient is adaptive to the time-stepping size of the training data; thus, it can achieve large time-stepping while keeping the accuracy. Numerical results show that the proposed method can learn accurate large time-stepping integrators for various differential equations, including chaotic Hamiltonian systems (the Henon-Heiles Model and the two-body problem, the double pendulum), dissipative pendulum, and the KdV equation.

**Questions:**

- What is the matrix $\bar{S}$ in Eq.(5)? Is it the same as the matrix $S$ in Figure 1? How to derive or approximate $S$ or $\bar{S}$ in applications?

- Line 263: is $u^{(i)}$ a typo for $u^{(i-1)}$? Is there any reason for selecting such random training data?

- Line 306+: the natation x is a typo of q. What if the system is stiff?

- KdV example: What is the training data, and how is it generated? How does the training data affect the performance of the proposed model?

**Ethical Concerns:**

["NO or VERY MINOR ethics concerns only"]

**Final Justification:**

This is a technically solid paper, presenting a robust method for learning a data-driven numerical integrator that incorporates energy conservation laws into the design of neural network approximations. The additional tests and comparison with the baselines further improve the paper. Thus, I keep my rating.

**Limitations:**

yes

**Quality:**

4

**Strengths And Weaknesses:**

Strengths:
- The paper proposes a robust method for learning a data-driven numerical integrator that incorporates energy conservation laws into the design of neural network approximations.
- The paper validates the proposed approach in a diverse set of systems, including nonlinear Hamiltonian systems and dissipative systems.
- The paper is well-organized, and the main idea is clearly presented. Additionally, the paper provides a detailed description of the experiments and the code, ensuring the results are reproducible.

Weakness:
The paper is solid overall. The following are minor weakness points:
- Lack of statistical robustness tests. Since the proposed integrator is learned from data, it is important to discuss its dependence on training data, including data size and the methods used to generate training data (single or multiple trajectories, which can be interpreted as the distribution of the training data). Additionally, it is crucial to test the learned integrator with different initial conditions, not just at the end of the training trajectory. Then, report the mean and standard deviation of the errors (in time, as shown in the last plot of Figure 3) for each example.
- Missing computational time for the examples, except the first example. Computation efficiency is a major motivation of the proposed method over classical methods (which use a smaller time step size). However, the complex NN approximation will take more time to compute the implicit integrator. It is unclear if the proposed method is more efficient. Therefore, it is important to report computation time for each example.
- Theorem 3.1 and Theorem 4.2 are more like definitions or simple facts. They are not sufficient to be called a theorem. In particular, the last sentence of Theorem 4.2 is not clear enough to be a statement in a theorem.
- The KdV example lacks the details, and the results are rough.

---

> ### Author Rebuttal · Authors · 2025-07-31
>
> We sincerely thank the reviewer for the valuable comments and suggestions. We have carefully addressed the concerns raised regarding statistical robustness, computational efficiency, the clarity and rigor of the theorems, and the completeness of the KdV example. We believe the revisions and clarifications provided below improve the overall quality and rigor of the paper. We respond to each point in detail as follows.
>
> - **Lack of statistical robustness tests. Additionally, it is crucial to test the learned integrator with different initial conditions, not just at the end of the training trajectory. Then, report the mean and standard deviation of the errors for each example.**
>
> To address the concern, we conducted additional experiments on both the Hénon-Heiles and the double pendulum. The aim was to evaluate the impact of training data distribution, data size, and initial conditions on the performance of the learned integrator.
>
> **Hénon-Heiles system**: We used irregular training data {$((u^{(i-1)},h^{(i)}),u^{(i)}) \mid i=1,...,I$ }, where the elements of $u^{(i-1)}$ are randomly sampled from the domain $[-1, 1]$ and each step size $h^{(i)}$ is drawn independently from $[0, 0.5]$. Each $u^{(i)}$ represents the one-step solution computed from $u^{(i-1)}$ using step size $h^{(i)}$ with a high-order solver.
>
> We performed experiments using $100$, $1000$, and $5000$ training samples, respectively. The neural network consists of $10$ layers, each with $100$ width. We set the learning rate to $0.00001$ and trained the model for $30000$ iterations.
> For testing, we selected random initial conditions from $[-0.4, 0.4]^4$, including high-energy cases with total energy exceeding $1/6$. For each test trajectory, a fixed step size $h$ was randomly chosen from $[0, 0.5]$ and applied over 100 time steps.
>
> **double pendulum**: We generated training data as follows. First, we randomly sampled initial angles $(q_1, q_2)$ from the interval $[-0.8, 0.8]$, and set the initial momenta $(p_1, p_2)$ to zero. For each initial condition, we randomly chose a fixed step size $h$ from the interval $[0, 0.5]$, and integrated the system forward for 100 time steps using a high-order solver. This process was repeated to generate $500$ trajectories, each of $100$ steps. From the resulting collection of one-step pairs, we randomly shuffled and selected $100$, $1000$, and $5000$ samples to construct training datasets. The neural network consists of $10$ layers, each with $100$ width. We set the learning rate to $0.0001$ and trained the model for $30000$ iterations.
>
> During testing, we followed the same protocol as in training: each test trajectory starts from a randomly chosen initial condition, with angles in $[-0.8, 0.8]$ and momenta set to zero. A fixed step size $h \in [0, 0.5]$ was randomly selected per test and kept constant over 100 steps.
>
> | Problem       | 100 train data                              | 1000 train data                             | 5000 train data                             |
> |---------------|---------------------------------------------|---------------------------------------------|---------------------------------------------|
> | **Hénon-Heiles** |                                             |                                             |                                             |
> | - training loss   | $1.64 \times 10^{-8} \pm 6.09 \times 10^{-9}$ | $1.83 \times 10^{-8} \pm 2.66 \times 10^{-9}$ | $1.90 \times 10^{-8} \pm 1.30 \times 10^{-9}$ |
> | - test loss    | $3.43 \times 10^{-8} \pm 1.98 \times 10^{-8}$ | $5.29 \times 10^{-9} \pm 4.45 \times 10^{-9}$ | $1.58 \times 10^{-9} \pm 1.60 \times 10^{-9}$ |
> | **2-Pendulum** |                                             |                                             |                                             |
> | - training loss   | $2.09 \times 10^{-7} \pm 2.37 \times 10^{-7}$ | $3.44 \times 10^{-7} \pm 1.11 \times 10^{-7}$ | $3.09 \times 10^{-7} \pm 3.05 \times 10^{-8}$ |
> | - test loss    | $1.22 \times 10^{-5} \pm 1.64 \times 10^{-5}$ | $6.43 \times 10^{-8} \pm 1.78 \times 10^{-8}$ | $4.24 \times 10^{-8} \pm 3.52 \times 10^{-8}$ |
>
> The above table shows the average training and test losses with standard deviations for different training dataset sizes. Each result is computed based on 5 independent training runs with different random data. For both systems, the training loss consistently converges to a similar small value. However, the test loss improves as the number of training samples increases.
>
> - **Missing computational time for the examples, except the first example**.
>
> We conducted a re-evaluation of the computation time across different systems. The computation time is generally similar to that of the Hénon-Heiles problem reported in the submitted paper. In particular, the proposed method achieves better accuracy with less computational time than the fourth-order methods. For the KdV equation experiment, the network was made as small as possible without sacrificing accuracy. The network used is quite small, yet the test loss was $1.706 \times 10^{-6}$, which is not much different from the test loss of $1.695 \times 10^{-6}$ when a larger network with 5 layers with width of 200 is used. For other experiments, the computation time can be further reduced by making the network smaller. We will include these results in the final version.
>
> | Problem        | Layer | Width | Steps | Proposed (sec) | 2 Order Methods (sec) | 4 Order Methods (sec) |
> |----------------|:-----:|:-----:|:-----:|----------------:|-------------------------:|-------------------------:|
> | Pendulum       |   5   |  50   |  200  |       0.579          |           0.503               |           1.137               |
> | Hénon-Heiles   |  10   | 100   |  300  |          2.090  |                   1.437  |                   3.533  |
> | 2-Body         |   5   |  50   |  300  |          2.862  |                   2.443 |                   7.693  |
> | 2-Pendulum     |   5   |  50   |  300  |          5.768  |                   5.197  |                  18.043  |
> | Duffing        |   5   |  50   |  300  |          1.020  |                    0.685  |                   1.084  |
> | KdV            |   2  | 30   | 3000  |        6.566  |                   1.634  |                  8.781  |
>
> - **Theorem 3.1 and Theorem 4.2 are more like definitions or simple facts. They are not sufficient to be called a theorem. In particular, the last sentence of Theorem 4.2 is not clear enough to be a statement in a theorem**.
>
> Thank you very much. We will correct the statements of these theorems appropriately.
>
> - **The KdV example lacks the details, and the results are rough**.
>
> We apologize for the lack of details. We consider this equation on the interval $[0,L], L=20$ under the periodic boundary condition. We used the training data with the time length of $T=30$ and the time step size of $\Delta t=0.01$. The spatial step size was $\Delta x=0.5$. The initial condition of the training data is constructed by superimposing two solitary wave solutions of the KdV equation: $u_0(x) = u_{sol}(x - x_1, c_1) + u_{sol}(x - x_2, c_2)$, where $u_{sol}(x, c) = \frac{1}{2}c \cdot \mathrm{sech}^2\left(\frac{1}{2}\sqrt{c} \cdot x \right)$ with $c_1=0.75, x_1=0.33L,c_2=0.4, x_2=0.65L$. The time evolution is computed by solving the KdV equation using the odeint solver with high accuracy, where spatial derivatives were approximated using finite difference methods. For testing, the computation was carried out from the time length $T=30$, with the start point being at the final state of the training dataset. We will include these details in the final version.
>
> - **What is the matrix $\bar{S}$ in Eq.(5)? Is it the same as the matrix $S$ in Figure 1? How to derive or approximate $S$ or $\bar{S}$ in applications?**
>
> $\overline{S}$ can be $S$ itself or an approximation of $S$ as far as $\overline{S}$ is skew-symmetric or negative-definite. A typical example where $S$ must be approximated is the KdV equation. In this equation, $S$ is the differential operator $\partial/\partial x$. Therefore, it is necessary to approximate this operator by, e.g., the finite difference method. Since the forward or backward difference does not result in a skew-symmetric matrix, the central difference method is usually used.
>
> - **Line 263: is $u^{(i)}$ a typo for $u^{(i-1)}$? Is there any reason for selecting such random training data?**
>
> Thank you very much. $u^{(i-1)}$ is indeed a typo. Yes, there is a reason behind selecting the training data randomly within the range [−1,1]. Our primary goal is to capture the characteristic chaotic behavior of the Hénon–Heiles system within the bounded region. This choice allows the training set to cover the region where trajectories remain bounded and exhibit complex, chaotic motion. Some of the data can have higher energy, providing information about the system's behavior near the boundary between bounded and unbounded motion.
>
> - **Line 306+: the notation x is a typo of q. What if the system is stiff?**
>
> Thank you very much. We will correct the typo. If the system is stiff, we should use the discrete gradient method for stiff problems as the base method. For example, the exponential AVF method (e.g., Wu et al., In Recent Developments in Structure-Preserving Algorithms for Oscillatory Differential Equations, Springer, 2018. Gu et al., arXiv:2110.04092, 2021)  is combined with the proposed approach.

---

> > ### Comment · Reviewer_gBGZ · 2025-08-05
> >
> > Thank the authors for the detailed responses. My concerns are addressed. The learning structured numerical integrators from data is a promising direction, and this paper proposes a robust method in this direction. In the revised version, the authors may want to explain the limitations of the learned integrators, particularly for those in the examples of Duffing and KdV, whose computational times are close to the 4th-order methods in the table.

---

### Note · Authors · 2025-08-15

We thank the Area Chair and Reviewers for their time and valuable feedback. We will include the important points from the discussions during  the review process and the rebuttal in the final version.

As far as we understand, the most important concern of the reviewers was the computation time, including the training time. To address this concern, we measured the computation time, including the training time, for various problems. As a result, the computation time of our method was typically between that of the second-order method and the fourth-order method. As shown in the paper, our method achieves higher accuracy than the fourth-order method. As pointed out by a reviewer, our method does not necessarily need to be used at all time steps. By using it only when higher computational accuracy is desired, computational time can be further reduced.

Another major concern was comparison with other machine learning methods, including PINNs. Because our method is a method for improving numerical integrators, its objective differs from methods that learn solutions to differential equations or the differential equations themselves from data. Therefore, fair comparison is difficult, but we conducted comparisons under experimental settings that were as fair as possible, and it was demonstrated that our method is also applicable to these tasks.

We hope that these explanations and the results of the numerical experiments have thoroughly addressed the reviewers' concerns.

We would like to again thank the Reviewers and the Area Chair for their time and commitment to the review process.

---

### Decision · Program_Chairs · 2025-09-17

**Decision:**

Accept (poster)

**Comment:**

This paper introduces a method for learning numerical integrators that preserve or dissipate energy, extending discrete gradient methods with neural parametrizations. The approach is theoretically grounded, with universal approximation results, and is validated on a diverse set of Hamiltonian, dissipative, and PDE systems. Reviewers noted that the paper is well written, reproducible, and technically sound, with potential impact in machine learning for the sciences.

The main concerns centered on computational cost and the scope of baseline comparisons. While some reviewers initially questioned efficiency claims and the relevance of comparisons to methods such as PINNs, COMET, and Stabilized Neural ODEs, the authors provided detailed timing analyses and additional experiments with PINN and time-reversal ODE baselines, demonstrating favorable trade-offs and strong performance. Clarifications were also given for the theoretical results, dissipative cases, and robustness to training data.

After rebuttal, two reviewers increased their scores, one maintained a positive evaluation, and one remained borderline reject, largely due to preferences for broader baseline coverage. Overall, I find that the strengths (including sound methodology, thorough experiments, and substantial author clarifications) outweigh the remaining concerns. Therefore, I recommend acceptance.